# Fast kernel-based association testing of non-linear genetic effects for biobank-scale data

Boyang Fu [1] ✉, Ali Pazokitoroudi [1], Mukund Sudarshan[2], Zhengtong Liu[1], Lakshminarayanan Subramanian[2,3] & Sriram Sankararaman [1,4,5] ✉

Our knowledge of non-linear genetic effects on complex traits remains limited, in part, due to the modest power to detect such effects. While kernel-based tests offer a versatile approach to test for non-linear relationships between sets of genetic variants and traits, current approaches cannot be applied to Biobank-scale datasets containing hundreds of thousands of individuals. We propose, FastKAST, a kernel-based approach that can test for non-linear effects of a set of variants on a quantitative trait. FastKAST provides calibrated hypothesis tests while enabling analysis of Biobank-scale datasets with hundreds of thousands of unrelated individuals from a homogeneous population. We apply FastKAST to 53 quantitative traits measured across ≈ 300 K unrelated white British individuals in the UK Biobank to detect sets of variants with non-linear effects at genome-wide significance.

Understanding the contribution of nonlinear genetic effects on complex traits is an important question in human genetics[1–7]. A powerful approach to identify such effects relies on grouping genetic variants into "sets" and testing their aggregated effect[8–13]. The mixed model framework offers a versatile approach to test such effects: capable of testing a wide range of linear and nonlinear relationships between genotype and trait by employing a kernel function that measures similarity between pairs of genotypes[11–13]. In practice, testing within the mixed model framework is computationally impractical for large datasets, so current approaches typically restrict their focus to linear additive models[11]. While biobank-scale datasets containing genetic and phenotypic data over hundreds of thousands of individuals provide the large sample sizes needed to identify nonlinear effects[14–16], computational challenges have limited these efforts.

We propose Fast nonlinear Kernel-based ASsociation Test (FastKAST), a scalable approach to test for nonlinear effects of a set of variants on a trait in a mixed model framework. Specifically, FastKAST permits the fitting of a wide class of kernel functions that model nonlinear effects of genetic variants on a trait (including the popular radial basis function (RBF) kernel). FastKAST combines a low-dimensional approximation to the kernel function[17] within a score test, obtaining calibrated $p$ values by fitting a distribution to genome-wide statistics obtained from a small number of permuted phenotypes[18]. As a result, FastKAST can efficiently test nonlinear associations in biobank-scale data.

Our theoretical and empirical analyses show that FastKAST provides calibrated hypothesis tests. Using extensive simulations across genetic architectures in which the phenotypes have a linear dependence on genotype (consistent with the known polygenic architecture of most complex phenotypes[19–21]) but no nonlinear dependencies, we find that FastKAST provides calibrated $p$ values. On small-scale datasets that permit exact computation, FastKAST is highly concordant with exact tests. To illustrate its utility, we applied FastKAST to 53 quantitative traits measured across $N \approx 300$K unrelated white British individuals in the UK Biobank (UKBB). Performing a genome-wide scan of nonlinear effects of genotypes measured on common SNPs with MAF ≥ 0.01 on the UKBB SNP array grouped into non-overlapping 100 kb windows, we found 75 windows with statistically significant nonlinear effects across 25 traits ($p < 3.27 \times 10^{-8} \left( \frac{0.05}{28,818 \times 53} \right)$ accounting for the number of sets and traits tested). To interrogate the nature of these effects, we repeated our analyses after regressing out pairwise interactions (in addition to linear effects) and on imputed genotypes in

[1]Department of Computer Science, UCLA, Los Angeles, CA, USA. [2]Department of Computer Science, Courant Institute of Mathematical Sciences, New York University, New York, NY, USA. [3]Department of Population Health, NYU Grossman School of Medicine, New York, NY, USA. [4]Department of Human Genetics, David Geffen School of Medicine, UCLA, Los Angeles, CA, USA. [5]Department of Computational Medicine, David Geffen School of Medicine, UCLA, Los Angeles, CA, USA. ✉e-mail: boyang1995@cs.ucla.edu; sriram@cs.ucla.edu

the UKBB to obtain eight significant associations across three quantitative traits (Alkaline phosphatase, Lipoprotein-A, and Urate) To further interpret the signals detected by FastKAST, we applied FastKAST to test for nonlinear effects in protein-coding genes across 53 quantitative traits. We detected 48 significant trait-gene pairs demonstrating nonlinear effects of which 35 overlapped with regions in our genome-wide scan. We observed 9/48 of the significant trait-gene pairs identified by FastKAST were not detected as significant using the linear model underlying SKAT[11]. We further compared FastKAST with the linear kernel in SKAT in the setting where we aim to identify either linear or nonlinear effects and observed that FastKAST has increased power in detecting significant trait-gene pairs compared to SKAT. Our results highlight the potential of FastKAST to uncover nonlinear genetic effects from Biobank-scale datasets.

## Results

### Methods overview

FastKAST tests for nonlinear effects between genotypes measured on a set of $M$ single nucleotide polymorphisms (SNPs) and a phenotype. The vector of phenotypes $y$, measured on $N$ individuals, is modeled as:

$$y \sim \mathcal{N}(X\beta, \sigma_g^2 K + \sigma_\epsilon^2 I_N)$$

Here $X$ denotes fixed effects. $K$ is a $N \times N$ kernel matrix obtained by applying a kernel function $k$ to every pair of genotypes over the $M$ SNPs to be tested, i.e., entry $(i, j)$ in the matrix $K$, $K_{i,j} = k(z_i, z_j)$ where $z_i$ ($z_j$) denotes the genotype of individual $i$ ($j$). $\sigma_g^2$ denotes the variance component associated with genetic effects while $\sigma_\epsilon^2$ denotes the variance component associated with residual effects. The kernel function can model different relationships between genotype and phenotype: the inner-product kernel $k(z_i, z_j) = z_i^T z_j$ implies a linear additive model, while the radial basis function (RBF) kernel $k(z_i, z_j) = exp(-\gamma \frac{\|z_i - z_j\|^2}{2})$ is a common kernel to model nonlinear relationships.

Testing for a genetic contribution in this model involves testing the null hypothesis: $\sigma_g^2 = 0$ which is commonly achieved using the score test[11]. While $p$ values for the score test can be computed efficiently when testing linear effects (as implemented in the SKAT software[11]), these approaches are computationally impractical for testing non-linear effects in large samples.

FastKAST approximates the kernel function by transforming the input genotypes to a randomized feature space[17] where the number of random features $D$ (termed the approximation dimension) determines the quality of the approximation. Combining the idea that an approximation dimension $D$ substantially smaller than the sample size $N$ is sufficient for approximating the kernel function with efficient linear algebra implementations allows FastKAST to efficiently compute $p$ values. While these $p$ value computations assume that the kernel hyperparameters are known (e.g., the $\gamma$ parameter for the RBF kernel), the more common setting is one in which the hyperparameter is unknown. In this more general setting (which is the setting that we focus on in this work), FastKAST adaptively selects the hyperparameter and obtains calibrated $p$ values by fitting a distribution to genome-wide statistics[18] (see Methods for details).

### Calibration of FastKAST

To assess the calibration of FastKAST, we performed simulations of quantitative traits with linear additive effects but no nonlinear effects. We simulated phenotypes based on whole-genome genotypes from unrelated white British individuals in the UK Biobank (UKBB) ($N = 337{,}205$ individuals and $M = 593{,}300$ SNPs on the UK Biobank Axiom array; see Methods for details on the dataset). We performed simulations under four genetic architectures: infinitesimal model (causal variant ratio = 1), where causal variant ratio refers to the proportion of variants that are causal to the outcome trait; non-infinitesimal model (causal variant ratio = 0.001) with a different range of minor allele frequencies (MAF) for the causal variants: [0.01, 0.05] (RARE), [0.05–0.5] (COMMON), [0.01, 0.5] (ALL). The trait heritability was set to $h^2 = 0.50$ in all settings.

We applied FastKAST with the radial basis function (RBF) kernel in non-overlapping 100 kb windows to a sub-sample of N = 50,000 individuals with phenotypes simulated above. We approximated the RBF kernel with approximation dimension $D = 50M$, where $M$ is the number of SNPs within each window. Since the goal of our work is to identify sets of SNPs with nonlinear effects, we need to first completely regress out the linear effect before testing for nonlinear effects. We observe that simply regressing out the effects of SNPs in the set being tested does not yield calibrated tests, likely due to correlation or linkage disequilibrium (LD) across SNPs (Supplementary Fig. 1). On the other hand, regressing out the linear effect within the target window as well as the additional neighboring windows can solve this issue (which we term a superwindow, the size of which is measured in multiples of the target window size). We empirically show that a superwindow of size five (e.g., target window plus two neighboring windows on each side) leads to calibrated $p$ values and appropriate control of the false positive rate (Supplementary Fig. 1). With this approach to control for linear effects, FastKAST obtains calibrated $p$ values across the architectures considered (Fig. 1 and Supplementary Table 1). While FastKAST adaptively chooses the kernel hyperparameter, our theory (Supplementary Note 1) and empirical results show that FastKAST remains calibrated even for a specific choice of hyperparameter (Supplementary Figs. 2, 3 and Supplementary Table 1).

### Power analysis of FastKAST

Our next experiment sought to compare the $p$ values obtained by FastKAST to an exact test. In the first set of experiments, we analyzed the correlation in $p$ values between an exact test using the RBF kernel with hyperparameter $\gamma = 0.1$ and the approximate kernel used by FastKAST in a simulation with causal variant ratio = 0.001 with $h^2 = 0.5$. We limited our sample size to 8000 individuals due to the limitations of computing the exact kernel. Since the approximation accuracy depends on the approximation dimension $D$, we explored the correlation between exact test $p$ values and FastKAST $p$ values by varying $D$. We found that values of $D \geq 50M$, where $M$ is the number of SNPs in the set, resulted in consistently high correlation ($\geq 0.9$) (Supplementary Fig. 4). More importantly, there is high concordance (98.7% across the settings tested) in the acceptance or rejection of the null hypothesis (at a significance level corresponding to what we employ in our real data analysis for a single trait of $p < \frac{0.05}{28{,}818}$) (Supplementary Fig. 5). To further validate our choice of the approximation dimension, we compared $p$ values from a test employing the exact kernel (RBF kernel with hyperparameter $\gamma = 0.1$) with FastKAST ($D = 50M$) on two real traits: Body mass index (BMI) and blood Mean Platelet Volume (MPV). Applying both tests to assess nonlinear effects within 100 kb windows across 5000 unrelated white British individuals, $p$ values obtained by FastKAST are highly correlated with those obtained by the exact test (Pearson correlation $\rho$ of 0.94 for both traits; Fig. 1b). These results remain consistent across values of the RBF kernel hyperparameter $\gamma$ and for different runs of FastKAST (Supplementary Figs. 6, 7).

We compared the statistical power of FastKAST relative to an exact test using simulated phenotypes with true nonlinear effects. We varied the RBF kernel hyperparameter $\gamma$ (a measure of the scale of nonlinearity), the kernel variance component $\sigma_g^2$ (a measure of the strength of the nonlinear signal), and the approximation dimension $D$ (with default values of $D = 50M$, $\gamma = 0.1$, and $\sigma_g^2 = 0.05$). For each setting, we randomly selected 2000 windows of length 100 kb across 5000 individuals and simulated phenotypes $y \sim \mathcal{N}(0, \sigma_g^2 K + \sigma_\epsilon^2 I)$, where $K$ is constructed using the RBF kernel with hyperparameter $\gamma$. Across these simulations, the power of FastKAST is indistinguishable from that of the exact test provided the approximation dimension $D \geq 50M$ (Fig. 1c). Based on these results,

we decided to use $D = 50M$ as our approximation dimension across the remaining experiments.

## Computational efficiency of FastKAST

We compared the runtime of FastKAST to the exact test with increasing sample size with the number of SNPs set to $M = 30$ (about the average number of SNPS in a 100 kb when analyzing SNPs from the UKBB genotyping array) and $D = 50M$. For each setting (a given set of $N$, $D$), we randomly subsampled $N$ individuals from UKBB and $M$ consecutive SNPs and reported the average runtime across 100 runs (10 replicates for sample sizes larger than 30K).

The exact test has a runtime that increases rapidly with sample size: requiring more than 5 h to analyze $N = 50K$ (Fig. 1d) and extrapolated to require over 100 days to analyze $N = 500K$ samples. On the other hand, even on the largest sample size of $N = 500K$ (with $M = 30$, $D = 50M$), FastKAST requires less than 5 min to analyze a single set (this includes the time to compute $p$ values across multiple hyperparameter

values and to analyze permuted phenotypes). We also found that the runtime of FastKAST scales quadratically as a function of the number of SNPs and approximation dimension, so that FastKAST is best suited for analyzing relatively small sets of SNPs (Supplementary Fig. 8).

## Application of FastKAST to identify nonlinear effects in the UK Biobank

We applied FastKAST to about 300K unrelated white British individuals in the UKBB. We tested non-overlapping 100 kb windows (considering SNPs with MAF > 1% in the UKBB genotyping array) to test for nonlinear effects using the RBF kernel and each of 53 quantitative traits (see Methods for details on data processing). For each window tested, we regressed out the linear effect of genotypes using a superwindow of size five. We included sex, age, and the top 20 genetic principal components (PCs) as covariates in all our analyses. We adopted a two-stage testing strategy. In the first stage, we used a small approximation dimension $D = 10M$ to efficiently test all trait-set pairs. We then

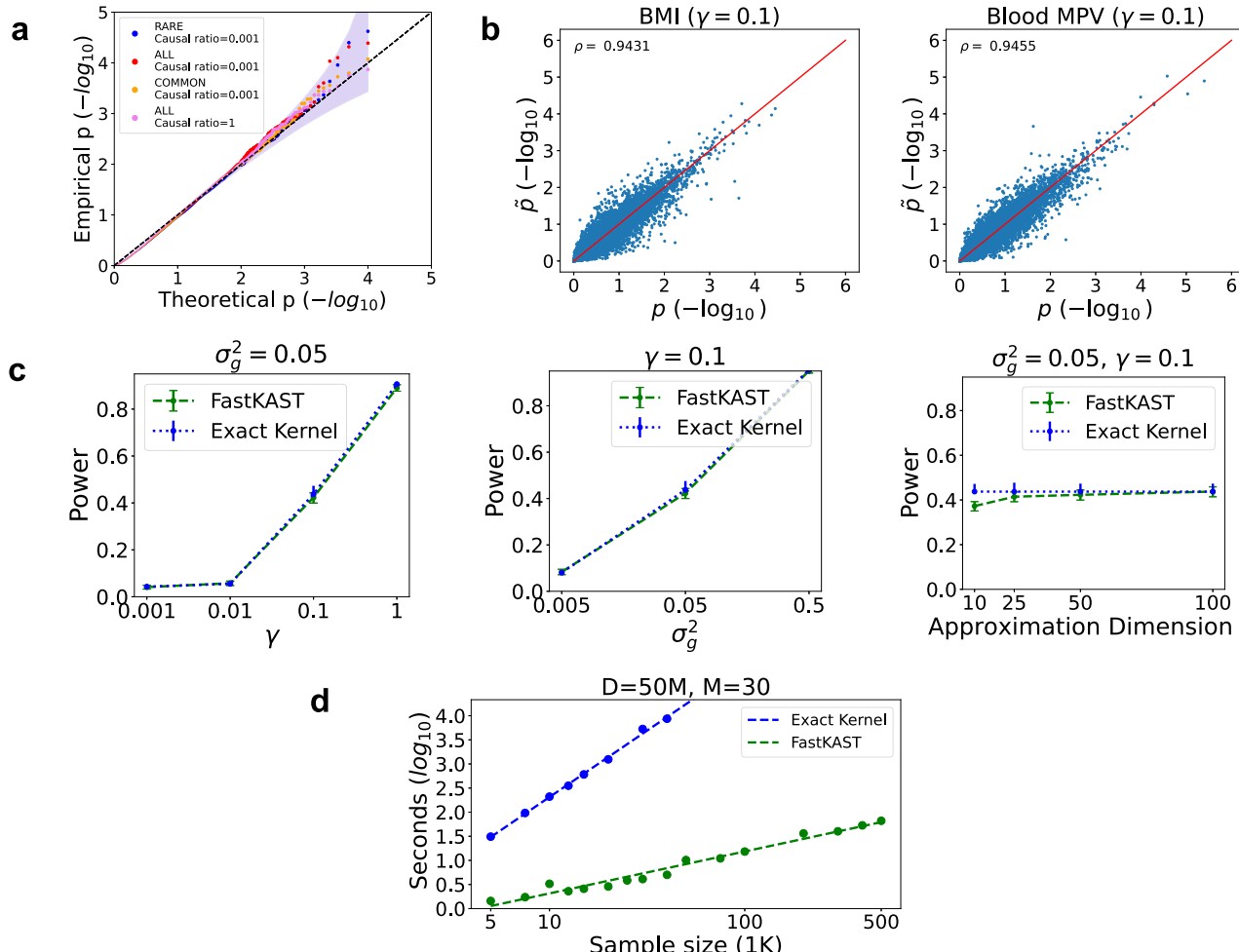

**Fig. 1 | Assessment of calibration, power, and scalability of FastKAST.**
**a** Calibration of FastKAST under null simulations that include linear effects but no nonlinear effects ($N = 50K$ individuals). We fixed SNP heritability at 0.50 while varying the ratio of causal variants ($\in \{0.001, 1\}$) and the range of minor allele frequencies (MAF) of the causal variants (ALL, COMMON, RARE). We applied FastKAST to test for nonlinear effects within 100 kb windows (after regressing out the linear effect in five windows centered around the tested window). The two-sided 95% confidence interval for the Q-Q plot was estimated using a beta distribution. **b** Comparison of $p$ values computed using FastKAST to an exact method. We analyzed Body mass index and Mean platelet volume (MPV) across 5000 unrelated white British individuals in the UK Biobank (UKBB). We tested each trait for nonlinear effects of SNPs in the UKBB genotyping array within non-overlapping 100 kb

windows using the exact RBF kernel and FastKAST (with approximation dimension $D = 50M$ where $M$ is the number of SNPs in a tested window and the kernel hyperparameter $\gamma = 0.1$). $p$ represents the $p$ value computed using the exact kernel; $\tilde{p}$ represents the $p$ value computed by FastKAST. **c** Power of FastKAST as a function of the kernel variance component $\sigma_g^2$, the kernel hyperparameter $\gamma$, and the approximation dimension $D$. We calculate the average (represented as a dot) across 2000 repetitions for each parameter setting and the bootstrap standard error bar across 1000 bootstrap replicates (denoted as a bar). **d** The runtime of FastKAST and the exact method as a function of sample size ($N$) for a fixed number of SNPs ($M = 30$) and approximation dimension $D = 50M$ (the default in this study). The exact method requires hours to analyze sample sizes larger than 50K. FastKAST remains efficient for sample sizes as large as 500K.

selected all the candidate trait-set pairs for which the estimated $p$ value passed a relaxed significance threshold ($\alpha = 1 \times 10^{-5}$). In the second stage, all the trait-set pairs were tested using a larger approximation dimension $D = 50M$, and only the trait-set pairs that are significant after Bonferroni correction for both the number of traits and sets tested ($p < \left(\frac{0.05}{28,818 \times 53}\right)$) across five different seeds were declared as significant. This two-stage approach drastically reduces the computational complexity compared to directly applying stage two across all trait-set pairs.

We detected 75 statistically significant associations ($p < 3.27 \times 10^{-8}$) for 25 traits (Supplementary Table 3). We further assessed the robustness of our results to population structure by varying the number of PCs included (from five to forty) and found the statistical significance to be numerically stable to the choice of the number of PCs (Supplementary Table 2).

We performed additional analyses to investigate the nature of the nonlinear effects at these loci. First, we repeated our analysis by regressing out linear and quadratic effects and repeated the test using FastKAST ("Nonlinear + non-quadratic" column in Supplementary Table 3). Previous studies have shown that apparent nonlinear genetic effects could potentially be explained by a model of linear effects involving untyped causal variants and correlation between tested genotypes with untyped causal variants[22]. We investigated this possibility by testing the significant loci using imputed genotypes (column "Nonlinear (imputed)" in Supplementary Table 3). We found that 24 out of the 75 trait-set pairs remained significant using imputed genotypes of which eight remained significant after removing both linear and quadratic effects.

Many of the regions with significant epistatic signals have been detected in previous association studies that typically focus on linear additive effects. The locus associated with MPV (12:122.3-122.4 Mb) overlaps the gene *CFAP251* which contains multiple variants strongly associated with platelet volume[23–25] as well as multiple rare variants associated with male fertility[26]. The region 6:160.9–161.0 Mb associated with Lipoprotein-A overlaps with the gene *LPA*, which has been shown to harbor multiple variants with a strong association with lipoprotein-related function[27–29] The locus 4:9.9–10.1 Mb associated with serum urate levels overlaps the solute transporter gene *SLC2A9* which harbors multiple variants associated with serum urate levels[30–33]. Variants in this gene have also been found to have sex-specific effects on urate levels[34]. To investigate potential sex-specific differences in effects on serum urate levels, we separately analyzed this locus in men and women. We computed $p$ values at the hyperparameter value that attained the minimum $p$ value ($\gamma = 0.1$) in men and women using FastKAST. We obtain a $p$ value of $6.8 \times 10^{-4}$ in men and $5.2 \times 10^{-7}$ in women even though the number of men and women is comparable in our analyses ($N = 132,020$ for men and $N = 150,496$ for women). Overall, these results suggest that many of the loci that we detect as showing strong evidence of nonlinear effects harbor variants with significant marginal additive effects.

### Comparison of FastKAST and SKAT to detect associations within protein-coding genes

To increase the interpretability of our findings, we next applied FastKAST to windows consisting of protein-coding genes. Restricting our analysis to protein-coding genes on autosomes and genes with at least three SNPs leads us to test 10,078 genes. We then applied the two-stage testing procedure as described in the previous section. We analyzed 10,078 genes and therefore defined the significance level as $\alpha = 0.05/(10,078 \times 53)$, where 53 is the number of traits tested. In the first experiment, we aimed to compare the tests for nonlinear effects as realized in FastKAST to a test for linear effects implemented in SKAT[11]. We applied FastKAST to windows defined by protein-coding genes. We also tested the same regions using SKAT with its default settings. FastKAST detected 48 trait-gene pairs as significant, with

## Significant association

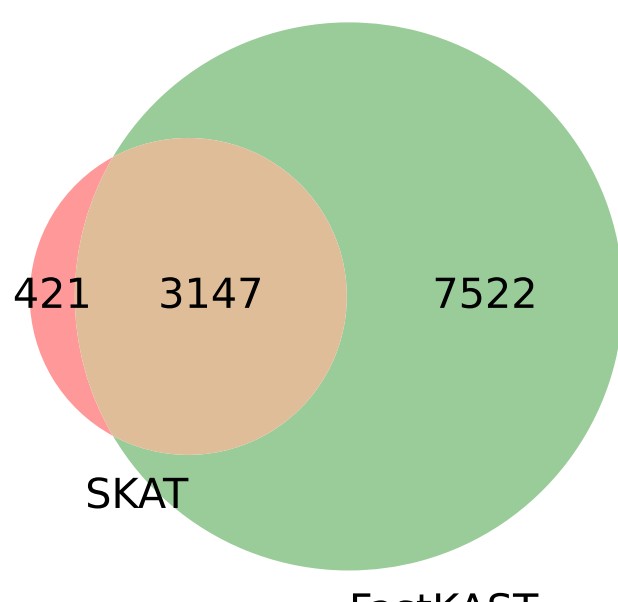

**Fig. 2 | Comparison of the significant hits discovered by SKAT and FastKAST when run on protein-coding genes.** The significance threshold corrects for multiple testing on the number of sets and traits tested, $\alpha = 0.05/(10,078 \times 53)$, where 10,078 is the total number of valid gene annotations and 53 is the total number of traits tested.

35/48 regions overlapping with the windows detected using the genome-wide scan. Among the 48 significant trait-gene pairs, nine were not detected as statistically significant when analyzed using SKAT (Supplementary Table 4).

To further understand the differences between FastKAST and SKAT, we applied both methods to the setting of general set-based association testing, i.e., to the setting in which we are interested in detecting either a linear or a nonlinear effect at a given set of variants. This setting contrasts with our previous analyses that focused on detecting nonlinear effects while regressing out linear effects. We applied both methods to the 53 quantitative traits with sets defined by the genetic variants in protein-coding genes. FastKAST was applied without removing linear effects as we wanted to understand the versatility of the test underlying FastKAST. Across all the traits, SKAT detected 3568 significant associations, of which 3147 were also detected by FastKAST. Additionally, FastKAST exclusively detected 7522 new association signals (Fig. 2 and Supplementary Table 5). Due to our application of FastKAST to test for either linear or nonlinear effects, we caution that these additional association signals may not all contain nonlinear effects but could instead represent regions harboring linear effects that were missed by SKAT.

## Discussion

We have described FastKAST, a computationally efficient algorithm that is capable of testing for nonlinear genetic effects in Biobank-scale data. FastKAST yields well-calibrated tests with little loss in power relative to an exact test. Applying FastKAST to common SNPs (MAF ≥ 0.01) on the UKBB genotyping array grouped into non-overlapping 100 kb windows and 53 quantitative traits measured across ≈300K unrelated white British individuals in UKBB, we discovered 75 nonlinear associations across 25 traits. We additionally analyzed these associations after regressing out pairwise interactions and on imputed genotypes in UKBB to find eight associations that

remain significant for Alkaline phosphatase, Lipoprotein-A, and Urate. We also applied FastKAST to the UKBB array SNPs grouped into protein-coding genes to detect 48 significant trait-gene pairs of which 35 associations overlapped with the associations detected in our genome-wide scan. In this setting, we also compared the results of FastKAST to those from the linear model underlying SKAT to find that 9/48 significant trait-gene pairs were not detected by SKAT. Finally, we compared FastKAST to SKAT in a general set-based association test setting that aims to detect either a linear or a non-linear association at a set of genetic variants and found 7522 trait gene-pairs that were detected as significant by FastKAST but not by SKAT while SKAT detected 421 trait-gene pairs that were missed by FastKAST.

We end with a discussion of the limitations of our approach and directions for future work. First, FastKAST is designed to analyze quantitative traits. FastKAST could potentially be extended to binary traits using a generalized linear mixed model with a logistic link function[11–13]. We leave a systematic evaluation of this extension to future work. Second, nonlinear interactions are represented in FastKAST using the class of shift-invariant kernels (which include the widely-used RBF kernel). In this work, all the experiments utilized the RBF kernel due to its popularity. FastKAST has the potential to be extended for a wider class of kernels (including kernels that are not shift-invariant) using other randomized approximations, e.g., Nyström kernel approximation[35]. For example, polynomial kernels might provide more interpretable insights into the basis of epistasis. In general, the optimal kernel remains unknown due to our limited understanding of the nature of epistasis. Indeed, the type of kernel that is appropriate is likely to depend on the trait and the set of genetic variants analyzed. We leave a more detailed exploration of alternative kernels and approximations for future work. Third, our results are localized to fairly broad windows of size 100 kb. While the application to disjoint windows of size 100 kb was motivated by computational and statistical considerations, we can apply FastKAST to other choices of windows. We have provided an alternative approach to define windows based on protein-coding genes annotation that can lead to more interpretable signals of epistasis. While the choice of windows could affect power when the underlying interaction effects are not confined to the window, we emphasize that FastKAST is consistently calibrated across all the null settings leading to high confidence in our signals. Fourth, we used only unrelated white British individuals across all our analyses. Analysis of related individuals or multiple ancestries will need to account for the possibility of population stratification (as is the case for most other analyses in the field). Further information from identity-by-descent (IBD), in addition to genetic ancestry, may be needed in these settings, which we leave for future work. Finally, we note that though we have shown strong evidence for the potential existence of higher-order feature interactions, our results must be interpreted with caution. The interpretation of genetic interactions is conditioned on the number and quality of SNPs analyzed. It has been shown that apparent interactions in the data can be explained by linearity with missing SNPs[3,22,36]. We have attempted to address this issue by replicating the loci discovered on the imputed genotypes in UKBB. While the imputed genotypes contain the vast majority of SNPs with minor allele frequency >1%, it is likely to be missing low-frequency SNPs. The availability of whole-exome and whole-genome sequencing data in the UK Biobank (and other biobanks) will allow a more thorough investigation of these effects.

## Methods

Let $y$ denote the vector of phenotypes measured on $N$ individuals and $Z$ denote the design matrix of genotypes over $M$ SNPs that are desired to be tested. The goal is to test for association between the set of $M$ SNPs and the phenotype.

We model the distribution of phenotypes, $y$, as:

$$y = X\beta + Z\alpha + \epsilon, \quad \alpha \sim \mathcal{N}(\mathbf{0}, \sigma_g^2 I_M), \quad \epsilon \sim \mathcal{N}(\mathbf{0}, \sigma_\epsilon^2 I_N) \qquad (1)$$

Here $y \in \mathbb{R}^N$, $X \in \mathbb{R}^{N \times P}$ denotes a matrix of covariates, $Z \in \mathbb{R}^{N \times M}$ is the design matrix of standardized genotypes measured over $M$ SNPs, and $\epsilon \in \mathbb{R}^N$ is the random vector of residual effects. $\beta \in \mathbb{R}^P$ is the vector of fixed effect coefficients while $\alpha \in \mathbb{R}^M$ is the vector of random effect coefficients. $\sigma_g^2, \sigma_\epsilon^2$ are the variance components associated with the genetic and residual effects. Integrating out the random effects, we have $y \sim \mathcal{N}(X\beta, \sigma_g^2 ZZ^\mathsf{T} + \sigma_\epsilon^2 I_N)$.

The above model assumes that the genotype has a linear and additive effect on phenotype. To model nonlinear effects, we transform the genotype using a nonlinear function $\phi : \mathbb{R}^M \to \mathbb{R}^Q$ leading to the following model:

$$y = X\beta + \Phi\alpha + \epsilon, \quad \alpha \sim \mathcal{N}(\mathbf{0}, \sigma_g^2 I_Q), \quad \epsilon \sim \mathcal{N}(\mathbf{0}, \sigma_\epsilon^2 I_N) \qquad (2)$$

Here $\Phi$ is the design matrix obtained by applying $\phi(z)$ to each individual genotype $z$. $\phi(z)$ is assumed to lie in a Hilbert space endowed with a reproducing kernel function $k(\cdot, \cdot)$[37]. Equivalently, we can write this model as:

$$y \sim \mathcal{N}(X\beta, \sigma_g^2 K + \sigma_\epsilon^2 I_N) \qquad (3)$$

Here $K$ is the $N \times N$ kernel matrix where $K_{i,j} = k(z_i, z_j)$, $i, j \in \{1, \dots, N\}$. For example, a common kernel is the radial basis function (RBF) kernel: $k(z_i, z_j) = \exp(-\gamma \frac{\|z_i - z_j\|^2}{2})$.

### Hypothesis testing

Testing for a genetic contribution to the phenotype in Equation (3) involves testing the null hypothesis $\sigma_g^2 = 0$. A commonly used approach to test the null hypothesis is the score test[11]. Under the null hypothesis, the score statistic $Q = \frac{1}{\sigma_\epsilon^2} y^\mathsf{T} P K P y$ is asymptotically distributed as a weighted sum of $\chi_1^2$ variables where the weights correspond to the eigenvalues of the matrix $PKP$ and $P = (I - X(X^\mathsf{T} X)^{-1} X^\mathsf{T})$ is the projection matrix. To compute the score statistic, an estimate of $\sigma_\epsilon^2$, typically the restricted maximum likelihood (REML) estimate, is used. More recent works[38,39] have characterized the sampling distribution of the score statistic in finite samples enabling the computation of exact $p$ values for the score test.

### Computation of $p$ values

A key challenge in computing $p$ values for the score statistic is the computation of all the eigenvalues of $PKP$. If we want to compute the $p$ values for the exact score test, we need to construct the kernel (time complexity depends on the type of kernel; $\mathcal{O}(N^2 M)$ complexity for the RBF kernel) followed by eigen-decomposition on the constructed matrix $K$ ($\mathcal{O}(N^3)$ time complexity). This approach is obviously infeasible for biobank-scale data.

Weighted linear kernels, *i.e.*, kernels of the form $K = ZWZ^\mathsf{T}$ where $W$ is a diagonal matrix with non-negative entries (used in popular software such as SKAT), permit efficient computation. However, these approaches are not applicable to kernels that model general nonlinear effects (like the RBF kernel), so that the computational complexity of testing for such effects scales as $\mathcal{O}(N^3)$.

### Random Fourier features

FastKAST relies on the observation that the kernel function can be approximated by mapping the input features to a randomized low-dimensional feature space[17]. For the class of shift-invariant kernel functions $k(z, z') = f(z - z')$ (for some function $f$) that include the popular RBF kernel, the kernel function can be approximated by projecting each input $z$ onto a random direction $\omega$ drawn from the Fourier

transform of $k$. Specifically, we approximate $k(\mathbf{z}, \mathbf{z}') \approx \tilde{k}(\mathbf{z}, \mathbf{z}') \equiv \tilde{\phi}(\mathbf{z})^T \tilde{\phi}(\mathbf{z}') = \frac{1}{D} \sum_{d=1}^{D} \tilde{\phi}(\boldsymbol{\omega}_d, b_d, \mathbf{z}) \tilde{\phi}(\boldsymbol{\omega}_d, b_d, \mathbf{z}')$, where $D$ denotes the number of random features (which we term the *approximation dimension*), $\boldsymbol{\omega}_d \in \mathbb{R}^M$, $b_d \in \mathbb{R}$, $\tilde{\phi}(\mathbf{z}) = \frac{1}{\sqrt{D}} [\phi(\boldsymbol{\omega}_1, b_1, \mathbf{z}), \ldots, \phi(\boldsymbol{\omega}_D, b_D, \mathbf{z})]$, $\boldsymbol{\omega}_d \overset{i.i.d}{\sim} p(\boldsymbol{\omega})$ where $p(\boldsymbol{\omega})$ denotes the Fourier transform of $k$, $b_d \overset{i.i.d}{\sim} Unif(0, 2\pi)$, and $\tilde{\phi}(\boldsymbol{\omega}, b, \mathbf{z}) = \sqrt{2} cos(\boldsymbol{\omega}^T \mathbf{z} + b)$. For example, in the RBF kernel with hyperparameter $\gamma = 1$: $k(\mathbf{z}, \mathbf{z}') = exp(-\frac{\|\mathbf{z} - \mathbf{z}'\|^2}{2})$, $p(\boldsymbol{\omega}) = (2\pi)^{-\frac{M}{2}} e^{-\frac{\|\boldsymbol{\omega}\|_2^2}{2}}$, and in this case $\boldsymbol{\omega}_d \overset{iid}{\sim} \mathcal{N}(\mathbf{0}, \boldsymbol{I}_M)$.

Given the $N \times D$ approximate feature matrix $\tilde{\boldsymbol{\Phi}} = [\tilde{\phi}(\mathbf{z}_1), \ldots, \tilde{\phi}(\mathbf{z}_N)]$, it follows that $\boldsymbol{K} \approx \tilde{\boldsymbol{\Phi}} \tilde{\boldsymbol{\Phi}}^T$. Prior work has shown that $\tilde{k}$ approximates $k$ for a sufficiently large number of features $D$[17] (we empirically explore the approximation dimension $D$ needed in our application). A key computational advantage of this approximation is that the approximate design matrix $\tilde{\boldsymbol{\Phi}}$ can be constructed in time linear in the sample size ($\mathcal{O}(NMD)$). We denote $\tilde{K} = \tilde{\boldsymbol{\Phi}} \tilde{\boldsymbol{\Phi}}^T$ as the approximate kernel matrix.

## Hypothesis testing with random Fourier features

We use a score statistic to test the null hypothesis that $\sigma_g^2 = 0$ using the random Fourier feature approximation to the kernel. Let $\tilde{Q} = \frac{1}{\sigma_e^2} \boldsymbol{y}^T \boldsymbol{P} \tilde{\boldsymbol{K}} \boldsymbol{P} \boldsymbol{y}$ denote the approximate score statistic where $\boldsymbol{P}$ is the projection matrix. We show that, under the null hypothesis, the approximate score statistic is distributed as $\sum_{n=1}^{N} \rho_n \chi_1^2$ where $\rho_n$ denotes the $n^{th}$ eigenvalue of the matrix $\boldsymbol{P} \tilde{\boldsymbol{K}} \boldsymbol{P}$ (Supplementary Note 1). We compute $\tilde{\boldsymbol{K}}$ using random Fourier features while we estimate the noise variance as $\hat{\sigma}_e^2 = \frac{\boldsymbol{y}^T \boldsymbol{P} \boldsymbol{y}}{N - P}$. Computing $p$ values for the approximate score statistic $\tilde{Q}$ requires computing the eigenvalues of $\boldsymbol{P} \tilde{\boldsymbol{K}} \boldsymbol{P}$ which can be computed from the SVD of $\boldsymbol{P} \tilde{\boldsymbol{\Phi}}$ with time complexity $\mathcal{O}(ND^2)$. Thus, the total time complexity of computing $p$ values using FastKAST is $\mathcal{O}(NMD + ND^2)$.

## Computing $p$ values when hyperparameters are unknown

Applying FastKAST typically requires choosing a value for the kernel hyperparameter $\gamma$. First, we note that the hypothesis test remains calibrated for any choice of hyperparameter. However, the choice of hyperparameter can influence power. A naive approach to perform hypothesis tests while integrating over choices of the hyperparameter would involve selecting a set of hyperparameter values $\{\gamma_1, \ldots, \gamma_H\}$ followed by computation of $p$ values $p_h$ for each hyperparameter $h$ (using the process described above). We then choose the minimum $p$ value: $p_* = min\{p_1, \ldots, p_H\}$ as the statistic. If the null hypotheses associated with the $H$ tests are all true, each of the $p$ values is calibrated under the null, and the $p$ values are independent, then it is well-known that $p_* \sim Beta(\alpha, \beta)$, where $\alpha = 1$ and $\beta = H$, i.e., the minimum of $H$ independent uniform random variables is distributed as a Beta random variable with density $f(x) = \frac{1}{H}(1-x)^{H-1}$. More generally, we can approximate the distribution of $p_*$ by a beta distribution whose parameters we estimate using one of two approaches.

1. Learn the distribution from observed data. In this approach, we fit a single beta distribution to the observed p-values to learn the parameters $\alpha$ and $\beta$. This approach assumes that the null hypothesis is true across most windows.
2. Learn the distribution from p-values computed from permuted phenotypes. In this approach, we fit a single beta distribution to the p-values computed on permuted phenotypes across all the windows. This approach relaxes the assumption that the null hypothesis is true across most windows.

We adopted the first approach for all the tests of epistasis on real data as well as in simulations where the signals of epistasis were assumed to be sparse. On the other hand, for the general association test on real data, we adopted the second approach by permuting ten times to generate the null distribution.

## Datasets

**Dataset used in simulations.** We obtained a set of $N = 337,205$ unrelated white British individuals measured at $M = 593,300$ common SNPs (MAF > 1%) genotyped on the UK Biobank Axiom array to use in simulations by extracting individuals that are >3rd-degree relatives and excluding individuals with putative sex chromosome aneuploidy. This dataset was used for all simulations except for Supplementary Table 1, which relied on the UKBB genotypes described below.

**UKBB genotypes.** For analysis of real traits, we restricted our analysis to SNPs that were presented in the UK Biobank Axiom array used to genotype the UK Biobank. SNPs with greater than 1% missingness and minor allele frequency smaller than 1% were removed. Moreover, SNPs that fail the Hardy–Weinberg test at significance threshold $10^{-7}$ were removed. We restricted our study to self-reported British white ancestry individuals, which are >3rd-degree relatives that is defined as pairs of individuals with kinship coefficient $<1/2^{(9/2)}$[40]. Furthermore, we removed individuals who are outliers for genotype heterozygosity and/or missingness. Finally, we obtained a set of $N = 291,273$ individuals and $M = 459,792$ SNPs to use in the real data analyses. We further excluded the MHC region in all our analyses (chr6: 25–35 Mb). For our analysis of fixed 100 kb windows, the number of SNPs per window has a mean of 17.5, a median of 16, and a range between 3 and 199. For our analysis of protein-coding genes, the distribution of the number of SNPs per gene has a mean of 15.6, a median of 7, and a range between 3 and 916. For both analyses, windows with SNPs number smaller than 3 were excluded from our analyses.

We also analyzed imputed genotypes across $N = 291,273$ unrelated white British individuals. We removed SNPs with greater than 1% missingness, minor allele frequency smaller than 1%, SNPs that fail the Hardy–Weinberg test at significance threshold $10^{-7}$, as well as SNPs that lie within the MHC region (Chr6: 25–35 Mb) to obtain 4,824,392 SNPs.

**Covariates and phenotypes.** We selected 53 quantitative traits in the UKBB, which we processed using inverse rank-normalization. We included sex, age, and the top 20 genetic principal components (PCs) as covariates in our analysis for all phenotypes. We used PCs computed in the UKBB from a superset of 488,295 individuals. Extra covariates were added for diastolic/systolic blood pressure (adjusted for cholesterol-lowering medication, blood pressure medication, insulin, hormone replacement therapy, and oral contraceptives).

## Reporting summary

Further information on research design is available in the Nature Portfolio Reporting Summary linked to this article.

## Data availability

The UK Biobank dataset used in this study is not publicly available but can be obtained by application (https://www.ukbiobank.ac.uk/).

## Code availability

FastKAST can be found at https://github.com/sriramlab/FastKAST with the required package installation script, exemplar simulation files, a script for running FastKAST, and results with tutorial analysis. The simulator used in the experiments can be found at https://github.com/alipazokit/simulator. SKAT (v.2.2.5) can be found at https://cran.r-project.org/web/packages/SKAT/index.html.

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

## Acknowledgements

This research was conducted using the UK Biobank Resource under application 33127. We thank the participants of the UK Biobank for making this work possible. This work was supported, in part, by NIH grants GM125055 (B.F., A.P., and S.S) and HG006399 (S.S.), and NSF grant CAREER-1943497 (B.F., A.P., and S.S.).

## Author contributions

B.F. was responsible for method development and experimental design. A.P. provided simulation scripts. M.S. and A.P. contributed to the initial project exploration. Z.L. assisted with experiments. L.S. provided suggestions and guidance. S.S. proposed the idea, secured funding, and provided project leadership, supervision, and guidance. B.F. and S.S. wrote the manuscript with input and feedback from all the authors.

## Competing interests

L.S. reports being the co-founder of Entrupy Inc, Gaius Networks Inc, and Velai Inc. and has also served as a consultant for the World Bank and the Governance Lab. The remaining authors declare no competing interests.
