## [Peer Review File · Nature Communications]

Fast Kernel-based Association Testing of non-linear genetic effects for Biobank-scale dataREVIEWER COMMENTS

Reviewer #1 (Remarks to the Author):

This paper addresses a potentially interest topic: statistical inference for non-linear effects in the genetic determination of quantitative traits. The paper as currently written has issues that need to be addressed:

Major issues:

1. The authors choose to focus on the analysis of genetic intervals along the genome (windows). There are clear computational and dimensionality advantages to the choice. However, there is no discussion of the possible loss of power associated with jointly analyzing many variants in each interval together, particularly if only one or a few variants in an interval is/are causal for a non-linear impact on phenotype, a model that surely is plausible. The sole focus on disjoint and exhaustive windows rather than considering windows based on genome annotation also seems a limitation.
2. While a primary strength of the method is computational efficiency, the higher-order relationship with number of markers and therefore the restriction to the analysis of a modest number of markers per window seems a serious limitation. If this interpretation is not correct, the opposite needs to be more clearly argued.
3. More details are required on method assumptions:
 - a. Must study participants be of a single ancestry or what is the impact of multiple ancestries?
 - b. Must study participants be unrelated or what is the impact if they are not?
 - c. The number of variants M in the interval assumedly varies by interval since the intervals are of fixed length. (How) do you deal with this?
 - d. The statement that the method can be extended to dichotomous traits should either be supported or softened. It certainly seems plausible.
 - e. The statement that the method can be applied to windows $<100\text{kb}$ seems quite important; can more explanation/justification be provided?
4. To me, correlations of exact and approximate $-\log_{10}$ pvalues of .90 or .94 is not convincing that the approximation is sufficiently accurate. More interesting is that 26 of 27 signals were captured, but that is a somewhat limited comparison.
5. I had a hard time understanding the UK Biobank application with the screening and follow-up stages. Assumedly this is done for computational efficiency. But doesn't this procedure preclude identifying non-linear effects in the absence of linear effects? I am sympathetic with the notion that this is the best place to look, but not as implied by the last sentence of the section that this is the only place to look. My apologies if I misread this.
6. Since the point is made that the method could be extended to other kernels, it would be good to say what the advantages of considering other kernels would be.
7. Please provide a bit more information on the software being made available and its generality with respect to the problem and methods described.

Minor issues:

8. The fact that the method as currently written is limited to quantitative traits should be in the title or at least the abstract.
9. The statement that understanding non-linear genetic effects on complex traits is a "central question" in human genetics is arguable; that it is an "important question" is clear. I personally would

go with the latter.

10. The statement that these approaches provide a powerful approach to identify non-linear effects does not seem obviously supported by the data. Perhaps a comparative statement would be better?

11. Do references 8 and 9 focus on non-linear effects? My recollection is not, but I did not go back to reread them.

12. The method would seem to handle any genetic variant, not just SNPs.

13. Are there are circumstances (other than simulation) in which the hyperparameter would ever be known?

14. You need to define "causal variants ratio." I think you mean the proportion of variants that are causal?

15. The statement that the standard score test is computationally infeasible for interaction effects would be helped by a reference.

16. Since the $O(NM^2)$ statement is said not to be relevant to kernels addressing interactions, I would drop that sentence.

17. How were the 30 UKB traits chosen?

Reviewer #2 (Remarks to the Author):

The authors proposed a computationally efficient approximated non-linear kernel test for nonlinear SNP-set effects on a phenotype. SNP set tests, such as kernel machine tests, have become popular in recent years especially for testing rare variant associations in whole genome/exome sequencing studies. Although kernel machine tests, such as SKAT, have been developed for allowing for testing linear and non-linear SNP effects in a set using linear and non-linear kernels, their computation using nonlinear kernels is intensive for large sample sizes. To overcome this, the authors proposed FastKAST, which approximates a nonlinear kernel function using a low dimensional randomized feature space constructed using the Fourier transformation of the kernel function. The authors performed simulation studies using small sample sizes to show the p-values calculated using FastKAST were similar to those calculated using the exact kernel function. They applied the method to sliding window analysis of several quantitative traits of the UK biobank GWAS data of ~300,000 independent samples and identified a few windows with significant nonlinear SNP set effects.

The proposed method FastKAST provides an interesting computationally scalable approach to non-linear SNP set association tests. A few comments:

- Although the proposed method is interesting and has potentials, the paper does not provide sufficient convincing evidence to support that non-linear SNP effects are practically real and the significance of the utility of the proposed method in the real world.
- The authors analyzed a small number of UKB traits. Given the large sample sizes in UKB and only a very small regions were found significant to have non-linear effects, one wonders whether these are false positive, how important non-linear SNP effects are in practice, and the practical value of detecting nonlinear effects. It would be useful if more convincing evidence can be provided to demonstrate non-linear effects are likely to be real and important in practice. It would also be useful to analyze more traits and robustly demonstrate the practical value of the method.
- The UKB analysis was performed using GWAS data instead of WGS data. As sliding windows were

used, the results may not have biological interpretation. It would be more interesting to demonstrate the benefits of FastKAST by applying it to rare variant analysis of WGS data and compare with the existing methods.

- The evidence of non-linear SNP effects is rather limited in the literature. The authors indicated that non-linear SNP effects could be due to linear effects involving untyped causal variants. For sequencing data, this will not be an issue, since all SNVs are sequenced. Hence this rationale of non-linear SNP effects will not be relevant. It would be useful to demonstrate non-linear SNP effects matter for WGS association studies, and FastKAST can significantly improve rare variant association tests by incorporating non-linear kernels.
- The authors focused on testing for non-linear SNP effects by adjusting for linear effects. It would be useful to also examine whether non-linear kernel based SNP set tests can improve the power of linear kernel based SNP-set tests for detecting SNP set effects, especially for rare variants.
- The authors did not evaluate the type I error rates of FastKAST. It would be useful if authors can perform simulation studies to show FastKAST controls the type I error rate at small significance levels, e.g., $\alpha=10^{-6}$.
- Figure 2 shows the p-values are exactly the same for multiple highly significant SNPs. This result looks a bit strange. Can the authors provide an explanation? Also, are these significant results biologically meaningful?
- The authors restricted analysis of independent samples in UKB. It would be useful to extend the method by including related samples.

#####

Reviewer 1:

This paper addresses a potentially interest topic: statistical inference for non-linear effects in the genetic determination of quantitative traits. The paper as currently written has issues that need to be addressed:

Major issues:

1. The authors choose to focus on the analysis of genetic intervals along the genome (windows). There are clear computational and dimensionality advantages to the choice. However, there is no discussion of the possible loss of power associated with jointly analyzing many variants in each interval together, particularly if only one or a few variants in an interval is/are causal for a non-linear impact on phenotype, a model that surely is plausible. The sole focus on disjoint and exhaustive windows rather than considering windows based on genome annotation also seems a limitation.

We thank the reviewer for raising this important point. The main focus in this work is to rigorously demonstrate the existence of non-linear genetic effects (i.e. to show that we can detect non-linear genetic effects at genome-wide significant levels while being calibrated) rather than to optimize the power to detect such effects. We view the choice of windows to examine as a choice that is external to FastKAST as FastKAST is designed to work with any set of input genomic regions (though this choice will impact power as the reviewer notes). We agree with the reviewer that the choice of genomic regions can impact the power to detect such effects. Choosing appropriate genomic regions will depend on the biological hypothesis being tested.

In the previous version, we adopted an unbiased approach (analogous to GWAS) to test for non-linear effects. This motivated our use of disjoint and exhaustive windows (Using a sliding window is feasible). In our revised manuscript, we have also applied FastKAST to an alternate set of genomic annotations defined by protein-coding genes. In this new analysis, we identified 45 robustly significant trait-gene pairs that demonstrate strong evidence for nonlinear effects (Table 2 of the revised manuscript that we excerpt below).

Table 2: **Protein-coding genes that yield significant evidence for non-linear effects.** Highlighted rows indicate whether the corresponding significant trait-gene pair region overlaps with the previous genome-wide analysis.

Trait	CHR	Gene	Start (Mb)	End (Mb)	$-\log_{10}(p)$
Age first birth	9	OLFML2A	127.5	127.6	9.8
Alanine aminotransferase	4	ZNF732	0.3	0.4	22.0
	22	PNPLA3	44.3	44.4	22.0
	22	SAMM50	44.4	44.5	22.0
Alcohol intake frequency	19	SPINT2	38.8	38.9	22.0
Apolipoprotein B	19	RELB	45.5	45.6	22.0
	19	APOE	45.4	45.5	22.0
	19	PVRL2	45.3	45.4	22.0
	19	MAU2	19.4	19.5	22.0
	19	BCAM	45.3	45.4	22.0
	19	NCAN	19.3	19.4	22.0
	19	TOMM40	45.4	45.5	22.0
Aspartate aminotransferase	22	PNPLA3	44.3	44.4	22.0
Cholesterol	19	NCAN	19.3	19.4	22.0
Creatinine	1	CASP9	15.8	15.9	22.0
	14	GSTZ1	77.8	77.9	22.0
Direct bilirubin	2	USP40	234.4	234.5	22.0
	2	DGKD	234.3	234.4	22.0
	2	UGT1A8	234.5	234.6	22.0
Eosinophil count	11	PRG3	57.1	57.2	22.0
	11	TNKS1BP1	57.1	57.2	22.0
HDL cholesterol	16	CETP	57.0	57.1	22.0
Lipoprotein-A	2	SESTD1	180.0	180.1	9.8
	6	LPA	161.0	161.1	22.0
	6	PLG	161.1	161.2	22.0
	6	SLC22A2	160.6	160.7	22.0
	6	IGF2R	160.4	160.5	22.0
	6	AGPAT4	161.6	161.7	22.0
	6	FNDC1	159.6	159.7	22.0
	6	SLC22A3	160.8	160.9	22.0
Mean corpuscular hemoglobin	6	LRRG16A	25.3	25.4	22.0
Mean platelet volume	1	TRIM58	248.0	248.1	22.0
	3	ARHGGEF3	56.8	56.9	22.0
	12	WDR66	122.4	122.5	22.0
	20	TUBB1	57.6	57.7	22.0
Mean sphered cell volume	1	SPTA1	158.6	158.7	22.0
Monocyte count	22	CECR1	17.7	17.8	22.0
Platelet count	1	C1orf150	247.7	247.8	22.0
Platelet distribution width	12	WDR66	122.4	122.5	22.0
	20	EDN3	57.9	58.0	22.0
	20	TUBB1	57.6	57.7	22.0
Total bilirubin	2	USP40	234.4	234.5	22.0
	2	UGT1A8	234.5	234.6	22.0
	2	DGKD	234.3	234.4	22.0
	2	TRPM8	234.8	234.9	22.0

We have now expanded on these issues in the Introduction Section, and have added a new section on the analyses of protein-coding genes titled “*Application of FastKAST to identify non-linear effects in protein-coding genes*” with the results summarized in Table 2 above, and in the Discussion section.

In the Discussion section, we now write:

“... Third, our results are localized to fairly broad windows of size 100 kb. *While the application to disjoint windows of size 100 kb was motivated by computational and statistical considerations, we can apply FastKAST to other choices of windows. We have provided an alternative approach to define windows based on protein-coding genes annotation that can lead to more interpretable signals of epistasis. While the choice of windows could affect power if the underlying interaction effects are not confined to the window, we emphasize that FastKAST is consistently calibrated across all the null settings leading to high confidence in our signals...*”

2. While a primary strength of the method is computational efficiency, the higher-order relationship with the number of markers and therefore the restriction to the analysis of a modest number of markers per window seems a serious limitation. If this interpretation is not correct, the opposite needs to be more clearly argued.

We thank the reviewer for bringing up this concern. We agree with the reviewer that the number of markers to be analyzed per window is indeed a limitation in our current implementation. However, we believe this is not a severe issue for two reasons. First, we have shown that our method can run on windows defined by protein-coding genes increasing the interpretability of our finding.. Second, even if the markers of interest are too large to be analyzed jointly, we can still partition them into small blocks and test each block separately. Though the power loss can be an issue depending on the search strategy, we want to emphasize that our method remains calibrated under the null regardless of the searching strategy. See Figure 1 (a) that we excerpt below (and Tables S2-S4 in the revised manuscript).

Figure 1: **Assessment of calibration, power and scalability of FastKAST.** (a) Calibration of FastKAST under null simulations that include linear effects but no non-linear effects ($N = 50K$ individuals). We fixed SNP heritability at 0.50 while varying the ratio of causal variants ($\in \{0.001, 1\}$) and the range of minor allele frequencies (MAF) of the causal variants (ALL, COMMON, RARE). We applied FastKAST to test for non-linear effects within 100 kb windows (after regressing out the linear effect in five windows centered around the tested window). (b) Comparison

3. More details are required on method assumptions:

a. Must study participants be of a single ancestry or what is the impact of multiple ancestries?

Population stratification will likely become an issue if multiple ancestries are involved in the sample (analogous to GWAS). Further, it is unclear whether these non-linear effects are similar across ancestries which can impact the power of tests to detect such effects. Thus, we recommend applying our method to individuals from a relatively homogeneous population. Further, we also recommend including genetic principal components (PCs) in our tests following the best practices from the GWAS literature. We addressed this limitation in the discussion section where we now write:

“Fourth, we used only the unrelated white British individuals across all our analyses. Analysis of related individuals or multiple ancestries will need to account for the possibility of population stratification (as is the case for most other analyses in the field). Further information from identity-by-descent (IBD) in addition to genetic ancestry may be needed in these settings which we leave for future work”

b. Must study participants be unrelated or what is the impact if they are not?

Our current analysis only guarantees calibration on unrelated individuals. It can be potentially applied to closely related individuals by incorporating the identity-by-descent (IBD) matrix as a fixed effect covariate in addition to the genotypic PC matrix. Further investigation is needed

regarding the calibration of this approach. We have added a statement in the discussion section about this limitation (see above).

c. The number of variants M in the interval assumedly varies by interval since the intervals are of fixed length. (How) do you deal with this?

Our results have shown that FastKAST's calibration is independent of the number of variants M included in each interval as shown in Figure 1(a) (above). Our current main analysis uses a fixed physical window -- a choice that we make to enhance interpretability. We can alternatively redefine the intervals to have an equal number of variants.

d. The statement that the method can be extended to dichotomous traits should either be supported or softened. It certainly seems plausible.

Testing on dichotomous traits can be achieved using the generalized link function but additional analyses will be needed to confirm the calibration and effectiveness of this approach. We have clarified this point in the abstract and the discussion section and have left the detailed analysis in the future.

"First, FastKAST is designed to analyze quantitative traits. FastKAST could potentially be extended to binary traits using a generalized linear mixed model with a logistic link function [11–13]. We leave a systematic evaluation of this extension to future work."

e. The statement that the method can be applied to windows $<100\text{kb}$ seems quite important; can more explanation/justification be provided?

The run-time complexity of the algorithm is a function of the feature size (number of SNPs) within the window. Therefore, the choice of 100Kb fixed window size is mainly due to our method's computation and memory requirement. We note that one can also define the window based on more interpretable boundaries, for example, functional annotations and this is a user-specified definition. To further explore alternative choices of windows, we redid our analysis defining windows based on known protein-coding genes. The detailed analysis and results can be found in new sections in the main text titled "*Application of FastKast to identify non-linear effects in protein-coding genes*" and "*Comparison of FastKast and SKAT*". We also discuss these choices in the Discussion section (see response to the first comment).

4. To me, correlations of exact and approximate $-\log_{10}$ p-values of .90 or .94 is not convincing that the approximation is sufficiently accurate. More interesting is that 26 of 27 signals were captured, but that is a somewhat limited comparison.

We thank the reviewer for this comment. We want to note that we computed the Pearson correlation of exact vs approximation kernel P-values (not the $-\log_{10}$ P-values) even though we plotted the scatter plot in the $-\log_{10}$ scale to highlight the accuracy for the extreme p-values approximations. This demonstrates the accuracy of p-value estimates across the range of values.

We have now added a new analysis where we compare the rejections of the approximate kernel to that of an exact kernel at a p-value threshold that corresponds to what we use in our data analysis in simulations (Supplementary Figure S8 that we excerpt below). The concordance in

the decisions (98.7% averaged across all settings tested) between the exact and approximate kernel (FastKAST) further demonstrate the accuracy of our algorithm. We now describe this analysis in the section titled “*Power analysis of FastKAST*”.

Figure S8: **Comparison of the power of FastKAST and a test that uses the exact kernel.** We compared the power of FastKAST to an exact kernel under simulations described in the **Power analysis of FastKAST** section. TP means the *True Positive*, FN means the *False Negative*. The first three settings correspond to a fixed value of the kernel hyperparameter γ while we vary the strength of the non-linear genetic effect σ^2 . The next three settings explore varying kernel hyperparameters γ while the non-linear genetic effects σ^2 is set to 0.05. The significance threshold used is the same as that of the UKBB analysis: $\frac{0.05}{28,818}$, where 28,818 is the total number of sets tested using 100kb window. The percentage of tests where the two tests agree across the settings is given by: $(\gamma = 0.1, \sigma_g^2 = 0.005) : 100\%$; $(\gamma = 0.1, \sigma_g^2 = 0.05) : 99.4\%$; $(\gamma = 0.1, \sigma_g^2 = 0.5) : 98.35\%$; $(\gamma = 0.01, \sigma_g^2 = 0.05) : 100\%$; $(\gamma = 0.001, \sigma_g^2 = 0.05) : 100\%$; $(\gamma = 1, \sigma_g^2 = 0.05) : 95\%$. “Positive” represents the number of significant hits a method yields while “Negative” represents the number of the tests that do not pass the significant threshold.

5. I had a hard time understanding the UK Biobank application with the screening and follow-up stages. Assumedly this is done for computational efficiency. But doesn't this procedure preclude identifying non-linear effects in the absence of linear effects? I am sympathetic with the notion

that this is the best place to look, but not as implied by the last sentence of the section that this is the only place to look. My apologies if I misread this.

We apologize for the lack of clarity in our description of this procedure. Our procedure is a two-step procedure that allows us to perform genome-wide scans (on non-overlapping windows) across many traits. In the first stage, we run our method with a smaller approximation dimension ($D=10M$) and identify windows that appear to be significant at a less stringent p-value threshold ($\alpha = 1 \times 10^{-5}$). We then test these windows with a larger approximation dimension ($D=50M$) reporting windows that have p-values below the Bonferroni threshold of all the windows tested. We have now clarified this procedure in the section titled “*Application of FastKAST to identify non-linear effects in the UK Biobank*”.

Our procedure can identify non-linear effects in the absence of linear effects. Indeed, in the revised manuscript, we include additional comparisons to SKAT (which is powered to detect linear effects) in a new section of the manuscript titled “*Comparison of FastKAST and SKAT*” and summarize in Figure 2 of the main text (excerpted below). To summarize, we compared the power of FastKAST to SKAT (which only tests for linear effects) on two exemplar traits (Mean corpuscular hemoglobin and BMI) with sets defined by protein-coding genes. Among all the valid tests for both methods, FastKAST identified 397 significant trait-set pairs relative to 202 identified by SKAT (with 181 shared associations) for Mean corpuscular hemoglobin, and FastKAST identified 303 significant trait-set pairs relative to 42 identified by SKAT (with 37 shared associations) for BMI. The above experiments demonstrate the power of using a non-linear kernel enabled FastKAST to detect additional loci relative to the use of a linear kernel.

Figure 2: Comparison of the significant hits discovered by SKAT and FastKAST on Mean Corpuscular Hemoglobin (left) and Body Mass Index (right) when run on protein-coding genes.

6. Since the point is made that the method could be extended to other kernels, it would be good to say what the advantages of considering other kernels would be.

We thank the reviewer for this suggestion. We chose to focus on the RBF kernel in this work because of its ability to model a wide range of non-linearities (interactions of all possible orders) which have led to its popularity in the statistics and machine learning literature. However, the RBF kernel has several limitations. The RBF kernel considers interactions of all possible orders. As a result, the RBF kernel might be less powerful at testing specific types of interactions for which alternative kernels might be better powered. For the same reason, the RBF kernel can

also be less interpretable relative to alternative kernels. We have now expanded on this point in the Discussion where we now write:

“Second, non-linear interactions are represented in FastKAST using the class of shift-invariant kernels (which include the widely-used RBF kernel). In this work, all the experiments utilized the RBF kernel due to its popularity. FastKAST has the potential to be extended for a wider class of kernels (including kernels that are not shift-invariant) using other randomized approximations, e.g., Nystrom kernel approximation [38]. For example, the polynomial kernel can provide more interpretable insights into the basis of epistasis. In general, the optimal kernel remains unknown due to our limited understanding of the nature of epistasis. Indeed, the type of kernel that is appropriate is likely to depend on the trait and the genetic variants analyzed. We leave a more detailed exploration of alternative kernels and approximations for the future work.”

7. Please provide a bit more information on the software is made available and its generality with respect to the problem and methods described.

We include all the necessary components needed to replicate our results. In our software (<https://github.com/sriramlab/FastKAST>), we detailed the IO requirements and have provided all the required hyperparameters and their descriptions.

We provide two versions of FastKAST for different purposes. Version 1 requires the user to specify the fixed physical window length and will automatically perform sequential window testing. Version 2 requires the user to provide an annotation file which allows the user to customize the window region.

We also provided an example genotype with the corresponding phenotype file and a jupyter notebook detailing how to perform the analysis.

Minor issues:

8. The fact that the method as currently written is limited to quantitative traits should be in the title or at least the abstract.

We have clarified this in the abstract which now reads:

“We propose, FastKAST, a Kernel-based approach that can test for non-linear effects of a set of variants on a quantitative trait.”

9. The statement that understanding non-linear genetic effects on complex traits is a “central question” in human genetics is arguable; that it is an “important question” is clear. I personally would go with the latter.

Fixed. The sentence now reads:

“Understanding the contribution of non-linear genetic effects on complex traits is an important question in human genetics.”

10. The statement that these approaches provide a powerful approach to identify non-linear effects does not seem obviously supported by the data. Perhaps a comparative statement would be better?

We also want to point out that our revised manuscript demonstrates that the kernel-based approach is significantly more powerful compared to the linear approach. Testing for

associations between protein-coding genes on an exemplar trait (Blood mean corpuscular hemoglobin), the existing linear kernel method (SKAT) detected 202 genes as significant while FastKAST detected 397 genes (of these 181 were shared across the two methods with 216 genes uniquely identified by the nonlinear kernel). We see qualitatively similar results on another trait: Body Mass Index (BMI). We have elaborated on these results in the Section “*Comparison of FastKAST and SKAT*” and in Figure 2 (see response to comment #5). Our results demonstrate the gain in power from using a nonlinear kernel for testing.

We have also changed “*powerful*” to “*versatile*” since kernel-based approaches allow for testing a wide variety of non-linear relationships by an appropriate choice of kernel.

“*The mixed model framework offers a versatile approach to test such effects:*”

11. Do references 8 and 9 focus on non-linear effects? My recollection is not, but I did not go back to reread them.

Correct. References 8 and 9 focused on testing (weighted) additive linear kernels which test for linear effects.

12. The method would seem to handle any genetic variant, not just SNPs.

Correct. The method is general and can be used to test other types of genetic variants (e.g. CNVs or STRs) or other genomic features (e.g. whether a gene carries a mutation).

13. Are there are circumstances (other than simulation) in which the hyperparameter would ever be known?

The hyperparameter is unknown in most of the settings we know. Thus, FastKAST adapts a grid-search approach to “learn” the best kernel for each set. We have shown in simulations that FastKAST is calibrated **both** in the setting where the hyperparameter is known and in the more realistic setting where the hyperparameter is unknown.

14. You need to define “causal variants ratio.” I think you mean the proportion of variants that are causal?

Correct. We have clarified this definition in the Section titled “*Calibration of FastKAST*”:

“*We performed simulations under four genetic architectures: infinitesimal model (causal variant ratio = 1) where causal variant ratio refers to the proportion of variants that are causal to the outcome trait*”

15. The statement that the standard score test is computationally infeasible for interaction effects would be helped by a reference.

We have now added text outlining the computational cost of the exact score test for non-linear effects in the Methods section on “*Computation of p-values*”:

“If we want to compute the p-values for the exact score test, we need to construct the kernel (time complexity depends on the type of kernel; $O(NM^2)$ complexity for the RBF kernel) followed by eigen-decomposition on the constructed matrix K ($O(N^3)$ time complexity).”

16. Since the O(NM2) statement is said not to be relevant to kernels addressing interactions, I would drop that sentence.

We have dropped the reference to the time complexity in that sentence.

17. How were the 30 UKB traits chosen?

In our revised manuscript, we now apply our method to 53 quantitative traits listed in Table S6 (see below). These traits were chosen to cover a diverse set of categories (anthropometric, blood biochemistry, bone, cardiovascular, eye, liver, and renal traits) and have been used in prior work from our group (Pazokitoroudi et al. *AJHG* 2021, Wu et al. *AJHG* 2022).

Category	Trait	Category	Trait
Anthropometry	BMD Heel T-score		LDL direct
	Basal metabolic rate		Lipoprotein-A
	Body mass index		Systolic blood pressure
	Height		Triglycerides
Blood biochemistry	Eosinophil count	Diabetes	Glucose
	High light scatter reticulocyte count		Hemoglobin A1c
	IGF-1	Eye	Corneal Hysteresis
	Lymphocyte count	Liver	Alanine aminotransferase
	Mean corpuscular hemoglobin		Albumin
	Mean platelet volume		Aspartate aminotransferase
	Mean spheroid cell volume		Direct bilirubin
	Monocyte count		Total bilirubin
	Platelet count	Other	Alcohol intake frequency
	Platelet distribution width		FEV1-FVC ratio
	RBC count		FVC
	RBC distribution width		Age first birth
	SHBG	Renal	Creatinine
	Testosterone		Creatinine in urine
	White blood cell count		Cystatin-C
	Bone	Alkaline phosphatase	
Calcium			Phosphate
Cardiovascular	Apolipoprotein A		Potassium in urine
	Apolipoprotein B		Sodium in urine
	C-reactive protein		Total protein
	Cholesterol		Urate
	Diastolic blood pressure		Urea
	HDL cholesterol		

Table S6: Traits analyzed in this work

#####

Reviewer 2:

A few comments:

- Although the proposed method is interesting and has potentials, the paper does not provide sufficient convincing evidence to support that non-linear SNP effects are practically real and the significance of the utility of the proposed method in the real world.

We thank the reviewer for pushing us to show the practical utility of our method. In our revised manuscript, we have performed several experiments to address the reviewer’s concerns:

1. We have applied FastKAST to 53 quantitative traits in UKBB to find 93 significant trait-loci pairs (where each locus is a 100 kb window along the genome and significance refers to loci that pass a stringent Bonferroni-corrected p-value threshold). Among these significant trait-loci pairs, 22 are also significant after removing the quadratic effect, 31 are also significant on the imputed genotype dataset, and 11 remain robustly significant across all three testings. We have now updated the section of the paper titled “Application of FastKAST to identify non-linear effects in the UK Biobank” with the results summarized in Table 1 of the main text that we reproduce below:

Table 1: Loci with statistically significant non-linear effects ($p \leq 1.47 \times 10^{-6}$ accounting for the number of sets tested). We report loci with p-values that are significant after regressing out linear effects of SNPs in five windows centered around the tested window. (The $-\log_{10}$ P-values are reported in entry *Array Non-linear*, with a precision level bounded by 22). We further report p-values after removing the quadratic effect of the current window in addition to the linear effect (The $-\log_{10}$ P-values are reported in *Array Non-linear + non-quadratic*, with a precision level bounded by 16). Further, we report the p-values after removing the linear effect when analyzing imputed genotypes in these windows (The $-\log_{10}$ P-values are reported in *Imputed Non-linear*, with a precision level bounded by 16). We highlight the loci which remain significant across all analyses.

Trait	Chr	Start (Mb)	End (Mb)	Array ($-\log_{10}(p)$)		Imputed
				Non-linear	Non-linear + non-quadratic	($-\log_{10}(p)$)
Alanine aminotransferase	22	44.3	44.4	22.0	0.5	2.9
Alkaline phosphatase	1	22.6	22.7	22.0	1.7	2.9
	6	24.1	24.2	22.0	0.6	0.4
	9	136.1	136.2	22.0	0.0	11.5
	9	136.2	136.3	22.0	8.0	12.6
	9	136.3	136.4	22.0	2.6	5.3
	19	49.1	49.2	22.0	5.3	9.1
Apolipoprotein B	19	19.3	19.4	22.0	8.4	7.3
	19	19.5	19.6	22.0	1.7	4.0
	19	19.6	19.7	22.0	0.9	5.5
	19	45.2	45.3	22.0	1.8	0.7
	19	45.3	45.4	22.0	0.0	8.9
	19	45.4	45.5	22.0	0.0	2.2
Aspartate aminotransferase	22	44.3	44.4	22.0	0.1	2.7
BMD Heel T-score	8	112.1	112.2	20.5	3.0	2.3
Body mass index	16	53.8	53.9	22.0	0.1	3.3
Cholesterol	19	19.3	19.4	22.0	4.6	5.4
	19	19.5	19.6	22.0	1.8	1.8
	19	19.6	19.7	22.0	0.3	3.6
Creatinine	1	15.8	15.9	22.0	0.1	5.0
Cystatin-C	20	23.9	24.0	22.0	2.6	0.4
Direct bilirubin	2	234.2	234.3	22.0	9.3	1.8
	2	234.3	234.4	22.0	2.6	9.3
	2	234.4	234.5	22.0	4.6	16.0
	2	234.5	234.6	22.0	2.2	16.0
	2	234.6	234.7	22.0	0.7	16.0
	2	234.8	234.9	22.0	2.7	2.4
	2	234.9	235.0	22.0	3.3	1.1
	11	2.9	3.0	22.0	0.3	1.3
Eosinophil count	11	57.1	57.2	22.0	1.1	0.6
HDL cholesterol	16	56.9	57.0	22.0	1.5	0.5
Hemoglobin A1c	6	20.6	20.7	22.0	0.0	4.0
IGF-1	11	30.6	30.7	22.0	4.8	0.9
LDL direct	19	19.3	19.4	22.0	5.6	5.9
	19	19.5	19.6	22.0	1.9	2.4
	19	19.6	19.7	22.0	0.3	3.8
	6	159.6	159.7	22.0	0.7	1.5
Lipoprotein-A	6	159.8	159.9	22.0	6.5	3.2
	6	159.9	160.0	22.0	5.9	4.4
	6	160.0	160.1	22.0	5.4	2.9
	6	160.2	160.3	22.0	10.1	4.0
	6	160.4	160.5	22.0	14.4	5.5
	6	160.5	160.6	22.0	5.7	13.1

	6	160.6	160.7	22.0	11.1	11.1
	6	160.7	160.8	22.0	16.0	14.2
	6	160.9	161.0	22.0	16.0	12.2
	6	161.0	161.1	22.0	16.0	8.6
	6	161.1	161.2	22.0	13.2	16.0
	6	161.2	161.3	22.0	10.8	5.4
	6	161.3	161.4	22.0	2.4	3.1
	6	161.5	161.6	22.0	9.5	3.1
	6	161.6	161.7	22.0	14.6	9.7
	6	161.7	161.8	22.0	5.1	3.0
	6	161.8	161.9	22.0	1.3	4.3
Mean corpuscular hemoglobin	6	135.4	135.5	22.0	0.4	8.1
	16	0.1	0.2	22.0	2.4	2.4
Mean platelet volume	3	56.8	56.9	22.0	1.2	10.4
	7	106.3	106.4	22.0	0.0	3.3
	12	122.1	122.2	22.0	7.9	14.7
	12	122.3	122.4	22.0	14.5	7.1
Mean spheroid cell volume	1	158.5	158.6	22.0	3.0	1.5
	6	140.7	140.8	22.0	8.2	0.8
Monocyte count	22	17.6	17.7	22.0	1.0	0.9
Platelet count	3	56.8	56.9	22.0	1.1	8.4
	6	135.4	135.5	22.0	0.0	3.4
Platelet distribution width	3	56.8	56.9	22.0	0.8	14.2
	20	57.5	57.6	22.0	0.3	1.3
	20	57.6	57.7	22.0	3.2	2.0
	20	57.8	57.9	22.0	1.2	3.7
RBC count	6	135.4	135.5	22.0	0.3	5.8
	9	136.1	136.2	22.0	0.1	1.5
SHBG	1	107.5	107.6	22.0	0.0	5.3
Total bilirubin	2	234.1	234.2	22.0	0.8	1.6
	2	234.2	234.3	22.0	15.2	4.1
	2	234.3	234.4	22.0	3.6	10.8
	2	234.4	234.5	22.0	6.8	16.0
	2	234.5	234.6	22.0	1.4	16.0
	2	234.6	234.7	22.0	0.6	16.0
	2	234.7	234.8	22.0	1.4	2.8
	2	234.8	234.9	22.0	1.8	4.9
	2	234.9	235.0	22.0	2.4	0.7
	2	235.0	235.1	22.0	2.1	2.9
Triglycerides	11	2.9	3.0	22.0	0.1	1.4
	11	117.1	117.2	22.0	2.5	0.2
Urate	1	15.8	15.9	22.0	0.4	5.5
	1	15.9	16.0	22.0	13.6	2.8
	4	9.8	9.9	22.0	0.4	2.1
	4	9.9	10.0	22.0	2.1	16.0
	4	10.0	10.1	22.0	16.0	16.0
	4	10.1	10.2	22.0	5.8	3.4
	4	10.2	10.3	22.0	0.8	2.5
	4	10.3	10.4	22.0	0.0	10.2
Urea	1	155.1	155.2	22.0	0.1	7.8

2. Additionally, we applied FastKAST to the same set of 53 traits but now focused on SNP sets limited to protein-coding genes. We detected 45 significant trait-gene pairs (significant after Bonferroni correction) which provide additional evidence for the presence of non-linear SNP effects. We have described these results in detail in a new section titled “*Application of FastKAST to identify non-linear effects in protein-coding genes*” with the results summarized in Table 2 that we reproduce below.

Table 2: **Protein-coding genes that yield significant evidence for non-linear effects.** Highlighted rows indicate whether the corresponding significant trait-gene pair region overlaps with the previous genome-wide analysis.

Trait	CHR	Gene	Start (Mb)	End (Mb)	$-\log_{10}(p)$
Age first birth	9	OLFML2A	127.5	127.6	9.8
Alanine aminotransferase	4	ZNF732	0.3	0.4	22.0
	22	PNPLA3	44.3	44.4	22.0
	22	SAMM50	44.4	44.5	22.0
Alcohol intake frequency	19	SPINT2	38.8	38.9	22.0
Apolipoprotein B	19	RELB	45.5	45.6	22.0
	19	APOE	45.4	45.5	22.0
	19	PVRL2	45.3	45.4	22.0
	19	MAU2	19.4	19.5	22.0
	19	BCAM	45.3	45.4	22.0
	19	NCAN	19.3	19.4	22.0
	19	TOMM40	45.4	45.5	22.0
Aspartate aminotransferase	22	PNPLA3	44.3	44.4	22.0
Cholesterol	19	NCAN	19.3	19.4	22.0
Creatinine	1	CASP9	15.8	15.9	22.0
	14	GSTZ1	77.8	77.9	22.0
Direct bilirubin	2	USP40	234.4	234.5	22.0
	2	DGKD	234.3	234.4	22.0
	2	UGT1A8	234.5	234.6	22.0
Eosinophil count	11	PRG3	57.1	57.2	22.0
	11	TNKS1BP1	57.1	57.2	22.0
HDL cholesterol	16	CETP	57.0	57.1	22.0
Lipoprotein-A	2	SESTD1	180.0	180.1	9.8
	6	LPA	161.0	161.1	22.0
	6	PLG	161.1	161.2	22.0
	6	SLC22A2	160.6	160.7	22.0
	6	IGF2R	160.4	160.5	22.0
	6	AGPAT4	161.6	161.7	22.0
	6	FNDCl	159.6	159.7	22.0
	6	SLC22A3	160.8	160.9	22.0
Mean corpuscular hemoglobin	6	LRRCl6A	25.3	25.4	22.0
Mean platelet volume	1	TRIM58	248.0	248.1	22.0
	3	ARRIGEF3	56.8	56.9	22.0
	12	WDR66	122.4	122.5	22.0
	20	TUBB1	57.6	57.7	22.0
Mean spheroid cell volume	1	SPTA1	158.6	158.7	22.0
Monocyte count	22	CECR1	17.7	17.8	22.0
Platelet count	1	C1orf150	247.7	247.8	22.0
Platelet distribution width	12	WDR66	122.4	122.5	22.0
	20	EDN3	57.9	58.0	22.0
	20	TUBB1	57.6	57.7	22.0
Total bilirubin	2	USP40	234.4	234.5	22.0
	2	UGT1A8	234.5	234.6	22.0
	2	DGKD	234.3	234.4	22.0
	2	TRPM8	234.8	234.9	22.0

3. Third, we compared the power of FastKAST to SKAT (which only tests for linear effects) on two exemplar traits (Mean corpuscular hemoglobin and BMI) when applied to sets defined by protein-coding genes. Among all the valid tests for both methods, FastKAST identified 397 significant trait-set pairs relative to 202 identified by SKAT (with 181 shared associations) for Mean corpuscular hemoglobin, and FastKAST identified 303 significant trait-set pairs relative to 42 identified by SKAT (with 37 shared associations) for BMI. The above experiments demonstrate the power of using a non-linear kernel enabled FastKAST to detect additional loci relative to the use of a linear kernel. We have now added these results in a new section to the manuscript titled “*Comparison of FastKAST and SKAT*” and summarize in Figure 2 of the main text (excerpted below):

Figure 2: Comparison of the significant hits discovered by SKAT and FastKAST on Mean Corpuscular Hemoglobin (left) and Body Mass Index (right) when run on protein-coding genes.

4. Finally, we would like to emphasize that a key challenge in detecting non-linear effects and one that we prioritize is to ensure that the false positive rate of our method is rigorously controlled (at a stringent p-value threshold). We show that FastKAST indeed controls the false positive rate both in theory and practice. As a result, we have high

confidence in the validity of the loci that we discover by application of FastKAST and we view these results as a starting point to investigate these effects in detail.

- The authors analyzed a small number of UKB traits. Given the large sample sizes in UKB and only very small regions were found significant to have non-linear effects, one wonders whether these are false positives, how important non-linear SNP effects are in practice, and the practical value of detecting nonlinear effects. It would be useful if more convincing evidence can be provided to demonstrate non-linear effects are likely to be real and important in practice. It would also be useful to analyze more traits and robustly demonstrate the practical value of the method.

We thank the reviewer for bringing up this question. To address the reviewer's concern, We have expanded our analyses to 53 quantitative traits in the UKBiobank and found 93 robustly significant trait-locus pairs. We have now updated the section of the paper titled "*Application of FastKAST to identify non-linear effects in the UK Biobank*" with the results summarized in Table 1 (see response to previous comment).

Further, we applied FastKAST to sets consisting of protein-coding genes for the same set of 53 traits to discover 45 significant trait-gene pairs. We have described these results in detail in a new section titled "*Application of FastKAST to identify non-linear effects in protein-coding genes*" with the results summarized in Table 2 (see response to previous comment).

- The UKB analysis was performed using GWAS data instead of WGS data. As sliding windows were used, the results may not have biological interpretation. It would be more interesting to demonstrate the benefits of FastKAST by applying it to rare variant analysis of WGS data and compare with the existing methods.

We note that our genome-wide analysis focused on applying FastKAST to non-overlapping 100kb windows (and not sliding windows as the reviewer suggests). While the application of FastKAST in this genome-wide setting has the advantage of discovering non-linear genetic effects in an unbiased manner (analogous to GWAS), the results are harder to interpret.

Following the reviewer's suggestion, we have now applied FastKAST to test the 53 quantitative traits with windows defined by the protein-coding genes. This analysis yielded 45 significant trait-gene pairs which demonstrated strong evidence of nonlinearity (20/45 of these trait-gene pairs were also found in the genome-wide analysis). To further explore these associations, we also ran SKAT (a test for linear relationships) on the protein-coding genes. Among these significant trait-gene pairs, 8/45 cannot be detected as significant using existing linear testing methods (SKAT) indicating the value of testing for non-linear effects.

We agree with the reviewer that it would be of high interest to apply this method to rare-variant analysis in WGS data. While we are in the process of applying FastKAST on WGS, more work needs to be done for QCing the WGS data. We view this as an important direction for future work which we now add to the Discussion:

“Finally, we note that though we have shown strong evidence for the potential existence of higher order feature interactions, our results must be interpreted with caution. The interpretation of genetic interactions is conditioned on the number and quality of SNPs analyzed. It has been shown that apparent interactions in the data can be explained by linearity with missing SNPs [3,22,39]. We have attempted to address this issue by replicating the loci discovered on the imputed genotypes in UKBB. While the imputed genotypes contain the vast majority of SNPs with minor allele frequency > 0.1%, it is likely to be missing rare SNPs. The availability of whole-exome and whole-genome sequencing data in the UK Biobank (and other biobanks) will allow a more thorough investigation of these effects. “

- The evidence of non-linear SNP effects is rather limited in the literature. The authors indicated that non-linear SNP effects could be due to linear effects involving untyped causal variants. For sequencing data, this will not be an issue, since all SNVs are sequenced. Hence this rationale of non-linear SNP effects will not be relevant. It would be useful to demonstrate non-linear SNP effects matter for WGS association studies, and FastKAST can significantly improve rare variant association tests by incorporating non-linear kernels.

To address the reviewer’s concern about the limited evidence of non-linear SNP effects, we have now run FastKAST on a larger set of 53 quantitative traits. Our updated results show strong non-linear effects for 93 trait-locus pairs.

We also now show that the use of a nonlinear kernel can improve association tests over the linear kernel. We applied FastKAST to our set of 53 quantitative traits focused on sets defined by protein-coding genes. We then applied SKAT to 45 significant trait-gene pairs to find that 8/45 of these pairs were not statistically significant with SKAT. We then compared FastKAST to SKAT for a single exemplar trait, Mean Corpuscular Hemoglobin. While SKAT detected 250 protein-coding genes as significant, FastKAST detected 400 genes of which 220 were unique to FastKAST. We have now added these results in a new section to the manuscript titled *“Comparison of FastKAST and SKAT”* and summarize in Figure 2 of the main text (see response to comment above).

- The authors focused on testing for non-linear SNP effects by adjusting for linear effects. It would be useful to also examine whether non-linear kernel based SNP set tests can improve the power of linear kernel based SNP-set tests for detecting SNP set effects, especially for rare variants.

We thank the reviewer for this suggestion. As mentioned above, we followed the reviewer’s suggestion and demonstrated the superior power of using non-linear kernel based testing compared to the linear counterpart even when testing relatively common genetic variants (as measured on the UKBB genotyping array). Given that the non-linear kernel already provides advantages over the linear kernel for relatively common variants, we hypothesize that these advantages will be more evident in the analysis of rare variants. Nevertheless, due to the technical challenges of QCing rare variants, we view this important direction as outside the scope of our current work.

• The authors did not evaluate the type I error rates of FastKAST. It would be useful if authors can perform simulation studies to show FastKAST controls the type I error rate at small significance levels, e.g., $\alpha=10^{-6}$.

We thank the reviewer for the concern about the type I error rate at a small significance level. We have now performed additional simulations. Averaging over 1.3 million tests, we show that FastKAST controls the type-I error at smaller significance levels like $\alpha = 1 \times 10^{-6}$ (we note that a formal statistical test shows that the p-values are never above the nominal significance level). We have included the relevant Table S4 below:

Superwindow	γ	α						λ_{gc}
		10^{-1}	10^{-2}	10^{-3}	10^{-4}	10^{-5}	10^{-6}	
5	0.001	1.006e-01	9.96e-03	1.01e-03	9.12e-05	6.90e-06	7.66e-07	1.01
5	0.010	1.003e-01	9.98e-03	9.64e-04	9.41e-05	4.59e-06	7.65e-07	1.01
5	0.100	1.001e-01	9.96e-03	1.01e-03	9.72e-05	9.95e-06	2.30e-06	1.01
5	1.000	9.97e-02	9.83e-03	9.44e-04	1.12e-04	1.15e-05	0.00e+00	1.00
5	10.000	1.003e-01	9.82e-03	9.70e-04	1.13e-04	1.30e-05	7.66e-07	1.01

Table S4: False positive rate for varying p-value thresholds and genomic inflation factor (aggregated) We report the false positive rate and genomic inflation factor at different p-value thresholds as we vary the hyperparameter and fixed the number of surrounding window size used to regress out linear effects (superwindow) to be 5. We reported false positive rates across all simulation settings resulting in a total of over 1.3 million valid tests. Among all tests, we do not observe any significance threshold at which the FPR is inflated significantly above the nominal level.

• Figure 2 shows the p-values are exactly the same for multiple highly significant SNPs. This result looks a bit strange. Can the authors provide an explanation? Also, are these significant results biologically meaningful?

The reason for the “exactly the same” p-value for multiple highly significant SNPs was due to the fact that the KDE p-value computation can only be reached to a certain level of precision (more specifically, $1e-22$). Any p-value smaller than that will yield 0 during the estimation. In the figure, for demonstration purposes, we will plot any p-value smaller than $1e-22$ as exactly $1e-22$.

• The authors restricted analysis of independent samples in UKB. It would be useful to extend the method by including related samples.

Extending FastKAST to related samples could be done by including the IBD or the relationship matrix included in the covariates. We view this extension as an important direction for future work as we mention in the Discussion:

“Fourth, we used only the unrelated white British individuals across all our analyses. Analysis of related individuals or multiple ancestries will need to account for the possibility of population stratification (as is the case for most other analyses in the field). Further information from identity-by-descent (IBD) in addition to genetic ancestry may be needed in these settings which we leave for future work.”

REVIEWER COMMENTS

Reviewer #1 (Remarks to the Author):

The revised manuscript remains of potential interest, but while the authors have addressed many of my concerns, some concerns remain.

1. I appreciate the authors placing relevant limitations in the discussion. However, the (current) limitations of the method to (a) quantitative traits on (b) unrelated individuals of (c) a single ancestry for (d) analysis only of non-rare variants data needs to be acknowledged early on, ideally in the abstract but at a minimum in the introduction.
2. "We found that values of $D \geq 50M$, where M is the number of SNPs in the set, resulted in consistently high correlation (≥ 0.9) (Fig S4)" is not particularly reassuring. The 1.3% concordance noted is more useful, but not entirely comforting given a 1.3% error rate for a genomes worth of tests.
3. "This two-stage approach drastically reduces the computational complexity comparing to directly applying stage two across all trait-set pairs." "Overall, these results indicate that the loci that show evidence of non-linear effects harbor variants with significant marginal effects." How many significant results does it miss? How can you make this conclusion?
4. Please explain the implications of the statement "We found that 11 out of the 93 trait-set pairs remained significant using imputed genotypes of which two loci remain significant after removing both linear and quadratic effects."
5. Please explain the pvalue threshold $p \leq 0.05/10079$.
6. Please explain the basis for the statement in bold: "Our experiments not only demonstrate the consistency our findings but also provide insight into the interpretability of the non-linear interactions."
7. Please explain the reason for using and comparing to SKAT if you are limiting yourself to variants with $MAF > 1\%$? In that case, should you (also) compare to the maximum among the corresponding single-variant tests?
8. FastKAST vs SKAT: "s a comparison, we also ran the same tests with SKAT. For MCH, after excluding the invalid tests for either SKAT or FastKAST, SKAT detected 202 significant trait-gene pairs compared to 397 for FastKAST with 181 shared signals (p – value $< 0.05/10,079$ for each method). The employment of a non-linear kernel enabled FastKAST to detect an additional 216 additional significant loci relative to a test for linear effects. For BMI, after excluding the invalid tests for either SKAT or FastKAST, SKAT detected 42 significant trait-gene pairs relative to 303 for FastKAST with 37 shared signals. For BMI, the employment of a non-linear kernel enabled FastKAST to detect an additional 266 additional significant loci relative to a test for linear effects (Figure 2, S9)." I find the MUCH larger number of associations identified allowing for non-linear effects for BMI and MCH in the UKBB highly surprising given the long-time common wisdom restated by the authors that interaction effects are difficult to identify. Please offer some greater intuition for this very surprising set of results which seem quite inconsistent with the subsequent statement in the discussion that "Applying FastKAST to 53 quantitative traits measured across $\approx 300K$ unrelated white British individuals in UKBB, we discovered 93 non-linear associations across 28 traits where 11 loci show strong evidence for higher-order non-linear effects." This comparison suggest a few more significant results per trait rather than the hundreds for MCH and BMI. I suspect I am misunderstanding the situation, but this needs to be clarified.

9. Minor comments:

- a. Gene names should be italicized.
- b. Windoes should be windows.

Reviewer #2 (Remarks to the Author):

In the revision, the authors did additional analyses of 53 phenotypes of the UK biobank data to study non-linear variant set effects. Besides genetic region analyses, they also added gene-level association tests, and compared the FastKAST results with linear effect based SKAT. These additional analyses are useful. However, some results are still worrisome. The evidence for the non-linear effects detected by FastKAST is still not sufficiently convincing.

- Genetic region analysis: All of the significant regions with non-linear effects reported in Table 1 have p-values $<10^{-22}$. No significant results in-between say p-value from 10^{-7} to 10^{-22} are reported. These findings are rather extreme and seem suspicious.

- Gene-based analysis: Similar concerns here. All the significant genes except for one have p-values $<10^{-22}$. Can the authors comment on whether these significant genes make sense biologically? It is a little surprising that no gene with non-linear effect was found to be associated with LDL, but found for other traits.

- o What is the distribution of the number of SNPs per gene? Did you use the genotyped SNPs or imputed SNPs? Do you have results for both for gene-level analysis?

- o Did you include the SNPs in 2KB upstream and downstream of each gene in your gene-level analysis? This will include the promoter of each gene in analysis.

- The authors compared the findings using FastKAST with those using SKAT for two traits: mean corpuscular hemoglobin and BMI, and showed FastKAST provided more significant findings. It would be useful if the authors can compare the findings for all 53 traits. So the readers won't feel the reported results are cherry-picking.

- In Figure 2, the authors demonstrated FaastKAST identified 216 associations that were not identified by linear-effect based SKAT for mean corpuscular hemoglobin. This suggests 216 genes/regions are likely to have non-linear effects. However, only two were listed in Table 1 and one was listed in Table 2. This is a bit confusing. One would expect more significant genes/regions with non-linear effects to show up in Tables 1 and 2 for mean corpuscular hemoglobin.

- The authors showed that among the 53 traits, FastKAST found 45 genes with significant non-linear effects at $\alpha=0.05/10076$. Given 53 traits were analyzed, one will need to control for multiple comparisons across traits. Hence $\alpha=0.05/(10,076*53)$.

- The authors applied SKAT to these 45 significant gene-trait pairs with non-linear effects and found 8/45 could not significant using linear-effect based SKAT. For a more fair comparison, it would be useful if the authors can provide a full comparison instead of focusing on these 45 gene-trait pairs identified by FastKAST. In particular, across the 53 traits, how many significant gene-trait pairs were identified by FastKAST, how many were identified by SKAT, and how many overlapped.

- The authors stated in their response that FastKAST controlled for error rates. However, the authors did not provide any type I error rate simulation results, e.g., at the type I error rate 10^{-6} . Such simulations are needed to evaluate whether type I error rate is properly controlled by FastKAST.

Reviewer #1 (Remarks to the Author):

The revised manuscript remains of potential interest, but while the authors have addressed many of my concerns, some concerns remain.

1. I appreciate the authors placing relevant limitations in the discussion. However, the (current) limitations of the method to (a) quantitative traits on (b) unrelated individuals of (c) a single ancestry for (d) analysis only of non-rare variants data needs to be acknowledged early on, ideally in the abstract but at a minimum in the introduction.

We have now added the characteristics of the data (quantitative traits, unrelated individuals from a homogeneous population) to the abstract and have pointed out that our analyses focus on common variants ($MAF \geq 0.01$) in the introduction.

2. “We found that values of $D \geq 50M$, where M is the number of SNPs in the set, resulted in consistently high correlation (≥ 0.9) (Fig S4)” is not particularly reassuring. The 1.3% concordance noted is more useful, but not entirely comforting given a 1.3% error rate for a genomes worth of tests.

The “98.7% concordance” (“1.3% concordance” stated by the reviewer) refers to the concordance of decisions (i.e. whether a hypothesis is accepted or rejected) based on p-values computed using the exact test versus the approximate test underlying FastKAST (Fig S8). Specifically, we find that across the 28,818 tests, there is a 1.3% mismatch in the decisions across the exact test and FastKast (i.e. cases where one approach would reject the null while the other accepts the null).

We agree with the reviewer that a 1.3% rate of false positives would be unignorable for genome-wide testing. In our experiments, a false positive refers to FastKAST rejecting the null hypothesis that a locus has either no genetic association with the trait or the true genetic effect of the locus on the trait is linear. We would like to emphasize that FastKast has a well-calibrated false positive rate as shown in our extensive simulation experiments (Figures 1 (a), Fig S2, Table S2). To further justify our argument, we performed additional experiments with different approximation dimensions, and demonstrated that FastKAST is robustly calibrated across different approximation dimensions (Fig S9).

On the other hand, the results on concordance relate to power so that the approximate and the true test might differ in their decisions and is expected in settings where power is modest (as is the case with genome-wide scans with stringent p-value thresholds). We view a 98.7% concordance is sufficient for these applications (and can be increased using larger approximating dimensions).

3. “This two-stage approach drastically reduces the computational complexity comparing to directly applying stage two across all trait-set pairs.” “Overall, these results indicate that the loci that show evidence of non-linear effects harbor variants with significant marginal effects.” How many significant results does it miss? How can you make this conclusion?

“How many significant results does it miss?”: We agree with the reviewer that our approach can miss true signals: using the prescreening strategy with a relatively low approximation dimension (D=10M) can potentially decrease the power of detecting the epistasis signal. To compensate for this loss, the strategy we applied here is to set a higher p-value threshold ($1e-5$) to allow more sets to pass to the second stage, where a higher approximation dimension (D=50M) will be used. Nevertheless, this two-stage strategy can miss true signals. For example, for mean corpuscular hemoglobin, we detected a total of three significant trait-set pairs with one-stage strategy, of which one of them in region CHR 22, 37.4-37.5 MB was missed in our updated two-stage strategy.

“How can you make this conclusion?”: This is simply based on the fact that most of the significant trait-set pairs which were detected to have significant epistatic effect also demonstrate significant linear (marginal) effect using SKAT (Table 2), and that the observation that many of the significant nonlinear trait-set pairs overlap or are adjacent to sites with significant GWAS signals found by previous studies. We have now modified this statement to: *“Overall, these results suggest that many of the loci that we detect as showing strong evidence of non-linear effects harbor variants with significant marginal effects.”*

4. Please explain the implications of the statement “We found that 11 out of the 93 trait-set pairs remained significant using imputed genotypes of which two loci remain significant after removing both linear and quadratic effects.”

We meant to say *“We found that 11 out of the 93 trait-set pairs remained significant using imputed genotypes, which also remained significant after removing both linear and quadratic effect”*. In the revised manuscript, we changed this statement to *“We found that 24 out of the 75 trait-set pairs remained significant using imputed genotypes of which eight loci remain significant after removing both linear and quadratic effects.”* The updated numbers are due to the changes in our approach for computing p-values and to the significance threshold (that now accounts for the number of traits tested in addition to the number of windows).

“Remove both linear and quadratic effect” was intended to answer the question: how many of the detected significant nonlinear trait-set pairs harbor higher-order interaction effects (beyond pairwise interactive effect); while “Using imputed genotype” was intended to answer the question: how many of the detected significant nonlinear trait-set pairs are more trustworthy and are less likely to be affected by missing causal variants.

We intended to highlight these eight trait-loci pairs, which demonstrate higher-order interaction effects across both array and imputed genotypes, as being of interest for follow-up analyses.

5. Please explain the pvalue threshold $p \leq 0.05/10079$.

The number 10,079 comes from the total number of genes we tested for each trait, which we discussed later in the “*Comparison of FastKAST and SKAT*” session. In our revised manuscript, we account for testing of multiple traits by making this threshold more stringent: $p < 0.05/(10,079 \times 53)$, where 53 is the total number of traits tested. We now clarify the choice of the p-value threshold in the text.

6. Please explain the basis for the statement in bold: “Our experiments not only demonstrate the consistency our findings but also provide insight into the interpretability of the non-linear interactions.”

While this statement was meant to imply that our finding of protein-coding genes with non-linear effects provides additional insights into the mechanisms underlying non-linear interactions, we realize that there remains additional work to be able to interpret the signals within protein-coding genes and have removed this sentence in our revised manuscript.

7. Please explain the reason for using and comparing to SKAT if you are limiting yourself to variants with MAF>1%? In that case, should you (also) compare to the maximum among the corresponding single-variant tests?

The purpose of the comparison with SKAT here is rather to demonstrate that FastKAST can detect additional genetic associations (including potential non-linear effects) when applied to sets of common variants compared to a linear model (SKAT). If rare variants are of interest, our method can easily adapt the weighting scheme used in SKAT to be able to prioritize these variants. In our current work, since both methods operate on the same set of input variants, we think that our experiments represent a fair comparison of the advantages of flexible models of genetic effects.

We agree with the reviewer that testing the maximum among the corresponding single-variant tests (or alternately, testing the minimum p-value) is an alternative method to compare to. However, such an approach brings with it additional issues. First, such an approach is likely to have reduced power since picking the minimum p-value or the maximum test statistic can only hope to approximately capture genetic signal at such loci. Second, we would need to devise strategies to compute the p-value of the maximum of single-variant tests where choice of such strategies would, in turn, impact power. Given the above complications, we have decided to leave these exploratory studies as future directions.

8. FastKAST vs SKAT: “s a comparison, we also ran the same tests with SKAT. For MCH, after excluding the invalid tests for either SKAT or FastKAST, SKAT detected 202 significant trait-gene pairs compared to 397 for FastKAST with 181 shared signals (p – value < 0.05 10,079 for each method). The employment of a non-linear kernel enabled FastKAST to detect an additional 216 additional significant loci relative to a test for linear effects. For BMI, after excluding the invalid tests for either SKAT or FastKAST, SKAT detected 42 significant trait-gene pairs relative to 303 for FastKAST with 37 shared signals. For BMI, the employment of a non-linear kernel enabled FastKAST to detect an additional 266 additional significant loci relative to a test for linear effects (Figure 2, S9).” I find the MUCH larger number of associations identified allowing for non-linear effects for BMI and MCH in the UKBB highly surprising given the long-time common wisdom restated by the authors that interaction effects are difficult to identify. Please offer some greater intuition for this very surprising set of results which seem quite inconsistent with the subsequent statement in the discussion that “Applying FastKAST to 53 quantitative traits measured across \approx 300K unrelated white

British individuals in UKBB, we discovered 93 non-linear associations across 28 traits where 11 loci show strong evidence for higher-order non-linear effects.” This comparison suggest a few more significant results per trait rather than the hundreds for MCH and BMI. I suspect I am misunderstanding the situation, but this needs to be clarified.

We apologize for the ambiguity in the description associated with these results. In general, we performed two different experiments to assess the utility of FastKAST. In the first set of experiments, we aim to explicitly test for non-linear effects at a region by applying FastKAST after regressing the linear effect in and around the target region. This strategy resulted in “93 non-linear associations across 28 traits” although the number has changed slightly in the revised manuscript (75 associations across 25 traits). A drawback of this approach is the potential loss in power to detect a non-linear effect when the linear effect is regressed out of the model. In the second set of experiments, we used FastKAST to detect any type of association (either linear or nonlinear) between SNPs within a genomic region and a trait and compared this approach to the linear-model based counterpart (SKAT). This experiment yielded about 70 genes associated with BMI for SKAT and a little more than 300 genes for FaskAST. In our revised manuscript, we applied this strategy across all 53 traits to find that SKAT detected 3,568 trait-set associations of which 3,149 were detected by FastKAST. Further, FastKAST identified another 7,524 new trait-set associations. Regions identified using the second strategy could contain non-linear effects or could simply represent regions with linear effects that were not detected by SKAT. Unlike in the first experiment where we regress out linear effects from the surrounding window, which also greatly penalized part of the epistatic effect, the second experiment has the potential to capture more linear and epistatic signals. We now provide additional explanations for the much higher number of signals detected in this experiment writing:

“To further understand the differences between FastKAST and SKAT, we applied both methods to the setting of general set-based association testing, i.e., to the setting in which we are interested in detecting either a linear or a non-linear effect at a given set of variants. This setting contrasts with our previous analyses that focused on detecting non-linear effects while regressing out linear effects. We applied both methods on the 53 quantitative traits with sets defined by the genetic variants in protein-coding genes. FastKAST was applied without removing linear effects as we wanted to understand the versatility of the test underlying FastKAST. As a comparison, we also ran the same tests with SKAT. Across all the tests, SKAT detected 3,568 significant associations of which 3,149 were also detected by FastKAST. Additionally, FastKAST exclusively detected 7,524 new association signals (Fig 2; Table S1). Due to the use of FastKAST to test for either linear or non-linear effects, we caution that these additional association signals may not all contain non-linear effects but could instead represent regions harboring linear effects that were missed by SKAT. Unlike in the previous experiment where we regress out linear effects from the windows surrounding the target window, which also greatly penalized part of the epistasis effect, this experiment has the potential to capture more linear and epistatic signals.”

9. Minor comments:

- a. Gene names should be italicized.
- b. Windoes should be windows.

Revised.

Reviewer #2 (Remarks to the Author):

In the revision, the authors did additional analyses of 53 phenotypes of the UK biobank data to study non-linear variant set effects. Besides genetic region analyses, they also added gene-level association tests, and compared the FastKAST results with linear effect based SKAT. These additional analyses are useful. However, some results are still worrisome. The evidence for the non-linear effects detected by FastKAST is still not sufficiently convincing.

- Genetic region analysis: All of the significant regions with non-linear effects reported in Table 1 have p-values $< 10^{-22}$. No significant results in-between say, p-value from 10^{-7} to 10^{-22} are reported. These findings are rather extreme and seem suspicious.

This behavior arose due to our approach to estimate p-values using the Gaussian kernel density estimation (KDE) to estimate p-values in the unknown hyperparameter setting which suffers from low precision at the tails. To address the above limitation arising from the KDE approach, we replaced it with beta distribution estimation. We described in detail how to use beta distribution for p-value estimation at the end of the section: **Computing p-values in the unknown hyperparameter setting**. After switching to the new strategy, the estimated p-values are calibrated in the tails (Figure 1a and Table S2 replicated below). The relevant Table 1 with updated p-values has been attached at the end of the responses for the reviewer to refer to.

Superwindow	γ	α						λ_{gc}
		10^{-1}	10^{-2}	10^{-3}	10^{-4}	10^{-5}	10^{-6}	
5	unknown	0.99	0.99	0.98	1.02	0.92	0.76	0.99

Table S2: **False positive rate for varying p-value thresholds and genomic inflation factor (aggregated)** We report the false positive rate and genomic inflation factor at different p-value thresholds assuming the hyperparameter is unknown (grid search hyperparameter γ from $\{0.001, 0.01, 0.1, 1, 10\}$) and fixed the number of surrounding window sizes using to regress out linear effects (superwindow) to be 5. We reported false positive rates ratio, defined by $\frac{FPR}{Threshold}$, across all simulation settings resulting in a total of over 1.3 million valid tests from the simulation setting (ALL, Causal ratio=1). Among all tests, we do not observe any significance threshold at which the FPR is inflated significantly above the nominal level.

- Gene-based analysis: Similar concerns here. All the significant genes except for one have p-values $<10^{-22}$. Can the authors comment on whether these significant genes make sense biologically? It is a little surprising that no gene with non-linear effect was found to be associated with LDL, but found for other traits.

We have resolved the concerns of the reviewer with the new approach to estimate p-values (see point above). Table 2 with the updated p-values has been attached at the end of our response.

The reviewer also noticed that LDL has no significant non-linear effect when testing protein-coding genes. We found that this was due to a numerical convergence issue for several loci across different traits, including LDL. When the p-value estimation did not converge, we excluded the corresponding test from the final results. To refine the algorithm, we further engineered the package and made it numerically more robust (increasing the maximum number of integration terms and adjusting the tolerance rate when computing p-values using the Davies' exact method). With these changes, we were actually able to discover several more significant epistasis regions for various traits, including LDL. After the software update, we found one trait-gene pair for LDL that is significant ($p < 0.05 / (10,078 \times 53)$). Please check the updated table 2.

What is the distribution of the number of SNPs per gene? Did you use the genotyped SNPs or imputed SNPs? Do you have results for both for gene-level analysis?

The distribution of SNPs number per set for the gene-annotation define window has a median of 7, mean of 15.6, and range between 3 and 916. The corresponding distribution of the fixed window test has a mean of 17.5, a median of 16, and a range between 3 and 199. In both tests, windows with SNPs number smaller than 3 were excluded from the calculation. We also added this information to the main text "**Datasets**" section.

All the experiments involving testing protein-coding genes were performed with genotyped SNPs. Based on the current results, we didn't further analyze this trait on the imputed dataset.

- Did you include the SNPs in 2KB upstream and downstream of each gene in your gene-level analysis? This will include the promoter of each gene in analysis.

We thank the reviewer for this suggestion. We did not include the 2KB upstream and downstream of each gene in our current test. Due to the time limitation, we decide to leave this exploration to future work.

- The authors compared the findings using FastKAST with those using SKAT for two traits: mean corpuscular hemoglobin and BMI, and showed FastKAST provided more significant findings. It would be useful if the authors can compare the findings for all 53 traits. So the readers won't feel the reported results are cherry-picking.

We followed the reviewer's suggestion and ran the comparison between FastKAST and SKAT on all 53 quantitative traits. Across all the traits tested, the significant trait-gene pairs detected by FastKAST overlap highly with those of the SKAT with 3,149 common signals. On the other hand, 419 trait-gene pairs were found exclusively by SKAT while 7,524 trait-gene pairs were detected by FastKAST but not by SKAT (Fig 2 that we replicate below and Table S1).

Figure 2: Comparison of the significant hits discovered by SKAT and FastKAST when run on protein-coding genes. The significance threshold corrects for multiple testing on the number of sets and traits tested, $\alpha = 0.05/(10,078 \times 53)$, where 10,078 is the total number of valid gene annotations and 53 is the total number of traits tested.

- In Figure 2, the authors demonstrated FaastKAST identified 216 associations that were not identified by linear-effect based SKAT for mean corpuscular hemoglobin. This suggests 216 genes/regions are likely to have non-linear effects. However, only two were listed in Table 1 and one was listed in Table 2. This is a bit confusing. One would expect more significant genes/regions with non-linear effects to show up in Tables 1 and 2 for mean corpuscular hemoglobin.

We apologize for the ambiguity in the description underlying these results. In general, we performed two different experiments to assess the utility of FastKAST. In the first set of experiments, we aim to explicitly test for a non-linear effect at a region. To achieve this, we excluded the main effect in the region surrounding the target region. A drawback of this approach is the potential loss in power to detect a non-linear effect when the linear effect is regressed out of the model. In the second set of experiments, we used FastKAST to detect any type of association (either linear or nonlinear) between SNPs within a genomic region and a trait and compared this approach to the linear-model based counterpart (SKAT). Regions identified using the second strategy could contain non-linear effects or could simply represent regions with linear effects that were not detected by SKAT. Unlike in the first experiment where we regress out linear effects from the surrounding window, which also greatly penalized part of the epistatic effect, the second experiment has the potential to capture more linear and epistatic signals

In our previous submission, approach 2 resulted in 216 genes that were detected by FastKAST and not by SKAT as being associated with MCH while approach 1 yielded 2 regions when applied across 100 kb window and one gene when applied to protein-coding genes (these numbers are 244, 2, and 2 respectively in our revised manuscript listed in Tables S1, 1, and 2).

We now provide additional explanations for these observations writing:

“To further understand the differences between FastKAST and SKAT, we applied both methods to the setting of general set-based association testing, i.e., to the setting in which we are interested in detecting either a linear or a non-linear effect at a given set of variants. This setting contrasts with our previous analyses that focused on detecting non-linear effects while regressing out linear effects. We applied both methods on the 53 quantitative traits with sets defined by the genetic variants in protein-coding genes. FastKAST was applied without removing linear effects as we wanted to understand the versatility of the test underlying FastKAST. As a comparison, we also ran the same tests with SKAT. Across all the tests, SKAT detected 3,568 significant associations of which 3,149 were also detected by FastKAST. Additionally, FastKAST exclusively detected 7,524 new association signals (Fig 2; Table S1). Due to the use of FastKAST to test for either linear or non-linear effects, we caution that these additional association signals may not all contain non-linear effects but could instead represent regions harboring linear effects that were missed by SKAT. Unlike in the first experiment where we regress out linear effects from the surrounding window, which also greatly penalized part of the epistatic effect, this experiment has the potential to capture more linear and epistatic signals“

- The authors showed that among the 53 traits, FastKAST found 45 genes with significant non-linear effects at $\alpha=0.05/10076$. Given 53 traits were analyzed, one will need to control for multiple comparisons across traits. Hence $\alpha=0.05/(10,076*53)$.

We followed the reviewer’s suggestion on adjusting the threshold by taking into account the multiple testing across traits. Currently, all the reported signals are chosen based on their p-values being lower than the threshold: $0.05/(\# \text{ sets tested per trait}) \times (\# \text{ traits tested})$

- The authors applied SKAT to these 45 significant gene-trait pairs with non-linear effects and found 8/45 could not significant using linear-effect based SKAT. For a more fair comparison, it would be useful if the authors can provide a full comparison instead of focusing on these 45 gene-trait pairs identified by FastKAST. In particular, across the 53 traits, how many significant gene-trait pairs were identified by FastKAST, how many were identified by SKAT, and how many overlapped.

We want to clarify that in the experiment that the reviewer referred to, our goal of comparing SKAT to FastKAST on the 45 significant trait-gene pairs with non-linear effects was to test if these regions also harbor linear effects. For this goal, comparing SKAT and FastKAST on those regions without significant non-linear signals would not provide additional information.

In the second experiment that we discussed above, our aim was to compare the power of FastKAST and its linear counterpart (SKAT). In this task, we followed the reviewer’s suggestion and ran the comparison between FastKAST and SKAT on all of the 53 quantitative traits. Across all the traits tested, the significant trait-gene pairs detected by FastKAST overlap highly with those of the SKAT with 3,149 common signals. On the other hand, 419 trait-gene pairs were found exclusively by SKAT while 7,524 trait-gene pairs were detected by FastKAST but not by SKAT (Fig 2 that we included in the response to comment above and Table S1).

• The authors stated in their response that FastKAST controlled for error rates. However, the authors did not provide any type I error rate simulation results, e.g., at the type I error rate 10^{-6} . Such simulations are needed to evaluate whether type I error rate is properly controlled by FastKAST.

We followed the reviewer’s suggestions to provide a higher precision for the type I error rate analysis. The detailed results can be found in Table S2 (shown below):

Superwindow	γ	α						λ_{gc}
		10^{-1}	10^{-2}	10^{-3}	10^{-4}	10^{-5}	10^{-6}	
5	unknown	0.99	0.99	0.98	1.02	0.92	0.76	0.99

Table S2: **False positive rate for varying p-value thresholds and genomic inflation factor (aggregated)** We report the false positive rate and genomic inflation factor at different p-value thresholds assuming the hyperparameter is unknown (grid search hyperparameter γ from $\{0.001, 0.01, 0.1, 1, 10\}$) and fixed the number of surrounding window sizes using to regress out linear effects (superwindow) to be 5. We reported false positive rates ratio, defined by $\frac{FPR}{Threshold}$, across all simulation settings resulting in a total of over 1.3 million valid tests from the simulation setting (ALL, Causal ratio=1). Among all tests, we do not observe any significance threshold at which the FPR is inflated significantly above the nominal level.

Table 1: **Loci with statistically significant non-linear effects** ($p < 3.27 \times 10^{-8}$ accounting for the number of sets and traits tested). We report loci with p-values that are significant after regressing out linear effects of SNPs in five windows centered around the tested window. (The $-\log_{10}$ p-values are reported in entry *Array Non-linear*, with a precision level bounded by 13). We further report p-values after removing the quadratic effect of the current window in addition to the linear effect (The $-\log_{10}$ p-values are reported in *Array Non-linear + non-quadratic*, with a precision level bounded by 13). Further, we report the p-values after removing the linear effect when analyzing imputed genotypes in these windows (The $-\log_{10}$ p-values are reported in *Imputed Non-linear*, with a precision level bounded by 13). We highlight the loci which remain significant across all analyses with ”***”; entries whose loci overlapped with those of the gene-set test are colored with grey background.

Trait	Chr	Start (Mb)	End (Mb)	Array ($-\log_{10}(p)$)		Imputed	
				Non-linear	Non-linear + non-quadratic	Non-linear ($-\log_{10}(p)$)	
Alanine aminotransferase	22	44.3	44.4	12.08	0.51	2.87	
Alkaline phosphatase	1	22.6	22.7	8.51	1.66	2.86	
	6	24.1	24.2	9.37	0.58	0.4	
	9	136.1	136.2	≥ 13	0.01	11.54	
***	9	136.2	136.3	10.55	8.0	12.57	
Apolipoprotein B	19	19.3	19.4	12.69	8.39	7.28	
	19	19.5	19.6	12.35	1.74	4.0	
	19	19.6	19.7	≥ 13	0.86	5.5	
	19	45.2	45.3	11.65	1.75	0.72	
	19	45.3	45.4	≥ 13	0.0	8.91	
	19	45.4	45.5	≥ 13	0.04	2.24	
Aspartate aminotransferase	22	44.3	44.4	9.48	0.08	2.74	
Body mass index	16	53.8	53.9	7.54	-0.3	3.29	
Cholesterol	19	19.6	19.7	11.87	0.27	3.55	
Creatinine	1	15.8	15.9	9.44	0.08	5.01	
Cystatin-C	20	23.9	24.0	≥ 13	2.63	0.4	
Direct bilirubin	2	234.2	234.3	8.66	9.27	1.8	
	2	234.3	234.4	≥ 13	2.59	9.26	
	2	234.4	234.5	11.98	4.61	≥ 13	
	2	234.5	234.6	≥ 13	2.21	≥ 13	
	2	234.6	234.7	≥ 13	0.67	≥ 13	
	2	234.9	235.0	≥ 13	3.29	1.13	
	11	57.1	57.2	10.81	1.12	0.64	
HDL cholesterol	16	56.9	57.0	≥ 13	1.52	0.55	
Hemoglobin A1c	6	20.6	20.7	8.21	0.01	3.99	
LDL direct	19	19.3	19.4	8.7	5.63	5.88	
	19	19.5	19.6	9.52	1.92	2.36	
	19	19.6	19.7	≥ 13	0.34	3.78	
Lipoprotein-A	6	160.2	160.3	≥ 13	10.08	4.02	
	6	160.4	160.5	≥ 13	≥ 13	5.52	
	6	160.5	160.6	≥ 13	5.66	≥ 13	
	***	6	160.6	160.7	≥ 13	11.08	11.06
	***	6	160.7	160.8	≥ 13	≥ 13	≥ 13
	***	6	160.9	161.0	≥ 13	≥ 13	12.2
	***	6	161.0	161.1	≥ 13	≥ 13	8.59
	***	6	161.1	161.2	≥ 13	≥ 13	≥ 13
	6	161.2	161.3	8.62	10.8	5.4	
	6	161.3	161.4	≥ 13	2.45	3.08	
	6	161.5	161.6	≥ 13	9.45	3.06	
***	6	161.6	161.7	≥ 13	≥ 13	9.71	
	6	161.7	161.8	11.64	5.06	3.02	
	6	161.8	161.9	≥ 13	1.28	4.28	

	6	159.8	159.9	9.41	6.46	3.18
	6	159.9	160.0	≥ 13	5.87	4.44
	6	160.0	160.1	≥ 13	5.41	2.89
Mean corpuscular hemoglobin	6	135.4	135.5	≥ 13	0.41	8.1
	16	0.1	0.2	12.69	2.45	2.41
Mean platelet volume	3	56.8	56.9	≥ 13	1.17	10.38
	12	122.3	122.4	8.96	≥ 13	7.1
Mean sphered cell volume	1	158.5	158.6	10.89	2.99	1.49
Monocyte count	22	17.6	17.7	10.16	1.02	0.87
Platelet count	3	56.8	56.9	11.73	1.12	8.41
Platelet distribution width	3	56.8	56.9	≥ 13	0.84	≥ 13
	20	57.5	57.6	≥ 13	0.29	1.25
RBC count	6	135.4	135.5	10.22	0.34	5.83
SHBG	1	107.5	107.6	8.77	0.03	5.35
Total bilirubin	2	234.1	234.2	9.19	0.81	1.57
	2	234.2	234.3	10.92	≥ 13	4.15
	2	234.3	234.4	≥ 13	3.6	10.81
	2	234.4	234.5	≥ 13	6.78	≥ 13
	2	234.5	234.6	≥ 13	1.39	≥ 13
	2	234.6	234.7	≥ 13	0.57	≥ 13
	2	234.7	234.8	9.42	1.36	2.83
	2	234.8	234.9	10.73	1.83	4.92
	2	234.9	235.0	≥ 13	2.39	0.74
	2	235.0	235.1	11.94	2.09	2.91
	11	2.9	3.0	8.86	0.08	1.38
Urate	1	15.8	15.9	10.43	0.44	5.5
	1	15.9	16.0	11.71	≥ 13	2.76
	4	9.9	10.0	≥ 13	2.07	≥ 13
***	4	10.0	10.1	≥ 13	≥ 13	≥ 13
	4	10.1	10.2	≥ 13	5.83	3.4
	4	10.2	10.3	≥ 13	0.81	2.47
	4	10.3	10.4	≥ 13	0.03	10.17
Urea	1	155.1	155.2	8.02	0.08	7.82

Table 2: **Application of SKAT to loci with statistically significant non-linear effects as detected by FastKAST.** We reported loci with p-values that are robustly significant after regressing out linear effects of SNPs in five windows centered around the tested window using FastKAST (denoted as $-\log_{10}$ Pval (FastKAST)), the precision is bounded by 13; and we also reported the corresponding $-\log_{10}$ p-value computed using SKAT (denoted as $-\log_{10}$ Pval (SKAT)), the precision is bounded by 315). Highlighted rows indicate whether the corresponding significant trait-gene pair region overlaps with the previous genome-wide analysis.

Trait	CHR	Gene	Start (Mb)	End (Mb)	$-\log_{10}$ Pval (FastKAST)	$-\log_{10}$ Pval (SKAT)
Alanine aminotransferase	22	PNPLA3	44.32	44.34	≥ 13	61.23
Alkaline phosphatase	9	C9orf96	136.24	136.27	≥ 13	282.96
Apolipoprotein B	19	NCAN	19.33	19.36	9.31	13.28
	19	MAU2	19.43	19.47	7.91	21.31
	19	ZNF101	19.78	19.79	10.33	8.20
	19	BCAM	45.31	45.32	≥ 13	132.82
	19	PVRL2	45.35	45.39	≥ 13	320.00
	19	APOE	45.41	45.41	≥ 13	320.00
Aspartate aminotransferase	22	PNPLA3	44.32	44.34	≥ 13	66.55
Creatinine	1	CASP9	15.82	15.85	8.62	0.06
	1	DNAJC16	15.86	15.89	8.45	1.52
Direct bilirubin	2	SAG	234.22	234.26	≥ 13	320.00
	2	DGKD	234.26	234.38	≥ 13	320.00
	2	UGT1A8	234.53	234.68	≥ 13	320.00
Eosinophil count	11	TNKS1BP1	57.07	57.09	≥ 13	2.37
	11	PRG3	57.14	57.15	≥ 13	5.69
HDL cholesterol	16	CETP	57.00	57.02	9.35	320.00
LDL direct	19	APOE	45.41	45.41	≥ 13	320.00
Lipoprotein-A	6	FNDC1	159.59	159.69	7.24	26.57
	6	IGF2R	160.39	160.53	≥ 13	320.00
	6	SLC22A2	160.64	160.68	7.06	248.48
	6	SLC22A3	160.77	160.87	≥ 13	320.00
	6	LPA	160.95	161.09	≥ 13	320.00
	6	PLG	161.12	161.17	12.68	295.99
	6	AGPAT4	161.56	161.65	≥ 13	320.00
Mean corpuscular hemoglobin	6	SCGN	25.65	25.70	8.54	34.88
	10	MSMB	51.55	51.56	7.4	1.26
Mean platelet volume	3	ARHGEF3	56.76	57.07	9.72	314.12
	12	WDR66	122.36	122.44	≥ 13	88.66
	20	TUBB1	57.59	57.60	≥ 13	141.04
Mean sphered cell volume	1	OR10Z1	158.58	158.58	9.07	11.10
	1	SPTA1	158.58	158.66	9.16	217.03
Platelet distribution width	12	WDR66	122.36	122.44	10.13	9.63
	20	TUBB1	57.59	57.60	≥ 13	320.00
	20	EDN3	57.88	57.90	11.53	20.29
RBC distribution width	19	APOE	45.41	45.41	10.59	93.73
SHBG	17	DNAH2	7.62	7.74	7.46	54.30
Total bilirubin	2	SAG	234.22	234.26	≥ 13	320.00
	2	DGKD	234.26	234.38	≥ 13	320.00
	2	UGT1A8	234.53	234.68	≥ 13	320.00
	11	SLC22A18	2.92	2.95	8.0	9.97
Urate	1	CELA2B	15.80	15.82	10.57	7.44
	1	CASP9	15.82	15.85	9.95	2.17
	1	DNAJC16	15.86	15.89	10.28	0.28
	4	SLC2A9	9.83	10.03	≥ 13	320.00
	4	WDR1	10.08	10.12	10.95	83.97
	4	MEPE	88.76	88.77	8.36	2.36
Urea	1	MUC1	155.16	155.16	7.47	2.69

REVIEWERS' COMMENTS

Reviewer #1 (Remarks to the Author):

The authors have addressed my remaining concerns.

Reviewer #2 (Remarks to the Author):

The authors made a good effort to address the comments provided in the previous reviews. The revision is improved. Many issues were addressed. There are a few remaining issues:

- The authors compared SKAT with FastKAST. They showed that the number of associations detected by both SKAT and FastKAST were 3,149, and FastKAST detected 7,524 new associations. The authors indicated that this did not mean 7,524 associations are non-linear. If you regress out linear effects, how many non-linear effects did FastKAST detect? If it is much smaller than 7,524, can you please provide explanations of the discrepancy?
- The type I error rate simulation results in Table S2 are confusing. The authors reported false positive rates. I am not sure how this is defined and how to benchmark them. It would be easier to understand the results if you can directly report the empirical type I error rates for different levels of the type I error rate alpha, e.g., $\alpha=10^{-4}$, 10^{-6} and 10^{-8} .
- Given the UKB whole genome sequencing data of $n=200,000$ samples are now available at RAP, if possible, it would be very useful if the authors can apply FastKAST to analysis of the UKB WGS data. This will significantly strengthen the paper.

Minor comment:

- Please use double space in the text. It will make the manuscript much easier to read.

REVIEWERS' COMMENTS

Reviewer #1 (Remarks to the Author):

The authors have addressed my remaining concerns.

We thank the reviewer for their valuable feedback, which significantly improved the quality of our work.

Reviewer #2 (Remarks to the Author):

The authors made a good effort to address the comments provided in the previous reviews. The revision is improved. Many issues were addressed. There are a few remaining issues:

- The authors compared SKAT with FastKAST. They showed that the number of associations detected by both SKAT and FastKAST were 3,149, and FastKAST detected 7,524 new associations. The authors indicated that this did not mean 7,524 associations are non-linear. If you regress out linear effects, how many non-linear effects did FastKAST detect? If it is much smaller than 7,524, can you please provide explanations of the discrepancy?

We appreciate the reviewer's insightful question. If we instead regress out the linear effects using our superwindow method, as shown in Table S4, we identified 48 significant trait-loci pairs exhibiting epistasis signals, including 9 that were not linearly significant when tested by SKAT. This discrepancy can be attributed to the fact that the general test that yielded 7,524 associations is intended to test for either a linear or non-linear effect. The substantially smaller number of windows that we detect after regressing out linear effects could be consistent with two possible explanations: 1) the number of regions in the genome that harbor non-linear effects is small, and many of these signals that the FastKAST general test detected were, in fact, additional linear effect associations that SKAT was unable to detect due to its lack of power; or 2) regions with non-linear effects tend to co-occur with regions harboring linear effects so that regressing out linear effects can also reduce the strength of the non-linear effect. In this work, we provide evidence for the existence of statistically significant and robust non-linear genetic effects for complex traits. Exploring whether non-linear effects could be more pervasive (beyond the highly significant signals that we detect) and testing which of models 1 vs 2 are consistent with our observations is an important direction for future work.

We have made one minor adjustment to the comparison statistics between FastKAST and SKAT (Figure 2). More specifically, for a small proportion of genes that span multiple regions so that the two methods are applied to each region, we report concordance only when both methods get the exact region (rather than our previous gene-level metrics). This resulted in a small change in the statistics: the number of regions detected by both methods dropped from 3,149 to 3,147; the regions found only by SKAT increased from 419 to 421, and regions found only by FastKAST decreased from 7,524 to 7,522.

- The type I error rate simulation results in Table S2 are confusing. The authors reported false positive rates. I am not sure how this is defined and how to benchmark them. It would be easier

to understand the results if you can directly report the empirical type I error rates for different levels of the type I error rate α , e.g., $\alpha=10^{-4}$, 10^{-6} and 10^{-8} .

We apologize for not making this part clearer. We reported the empirical type I error in Table S2, up to the precision of 10^{-6} , as the reviewer previously suggested. Previously, each entry of the column reported the ratio between the empirical type I rate for a given threshold α to α (so that an ideal method would be expected to be close to 1). To make it easier to understand, we have now updated table S1 (originally table S2) to report the actual type I error rate (FPR) for each value of α .

- Given the UKB whole genome sequencing data of $n=200,000$ samples are now available at RAP, if possible, it would be very useful if the authors can apply FastKAST to analysis of the UKB WGS data. This will significantly strengthen the paper.

We appreciate the reviewer's suggestion to apply FastKAST to the analysis of the UKB WGS data. We concur that this could significantly enhance the impact of our paper. However, given the current issues with access to the WGS data and careful QC needed, undertaking this task would demand a considerable commitment of time and resources. Nevertheless, we acknowledge the value of this suggestion and plan to explore it as a part of our future work

Minor comment:

- Please use double space in the text. It will make the manuscript much easier to read.

Done.